# Conserved genetic markers reveal widespread diatom sexual reproduction in the global ocean

Gust Bilcke [1,2,3,9], Lucia Campese [4,9], Rossella Annunziata[4], Luz Amadei Martínez [3], Camilla Borgonuovo[4], Nadine Rijsdijk [1,2,3], Peter Chaerle[3,5], Koen Van den Berge [6], Sofie D'hondt [3], Daniele Iudicone[4], Marina Montresor [4], Maria Immacolata Ferrante [4,7,10] ✉, Klaas Vandepoele [1,2,8,10] ✉ & Wim Vyverman [3,10] ✉

Sexual reproduction is a nearly universal characteristic of the eukaryotic life cycle, yet it is rarely observed in natural populations of micro-eukaryotes. Sex is particularly relevant for diatoms, a key group of marine and freshwater phytoplankton, where sexual reproduction counters a progressive cell size reduction due to cellular division. Here, we leveraged controlled sex transcriptome experiments of four diatom species to develop a robust method for in situ monitoring of sexual reproduction events. The resulting panel of conserved marker genes was validated for specificity and sensitivity using metatranscriptomic profiling of a natural estuarine community undergoing massive sexual reproduction of multiple species in response to increased salinity. Analysis of metatranscriptomic data linked with Metagenome-Assembled Genomes from the *Tara* Oceans expedition revealed widespread and coordinated expression of these markers across nine diatom genera, complemented by observations of sexual stages in automated imaging resources. Our results reveal that diatom sexual reproduction is more widespread in the global ocean than previously thought, encompassing both dominant bloom-forming species and rare taxa. Our panel of markers to detect sexual reproduction in natural environments paves the road to better understand the interplay between endogenous and environmental controls of this pivotal process, essential for the diatoms' evolutionary success.

Sexual reproduction is a nearly universal feature of the life cycle of eukaryotic organisms and is well-documented in animals and plants[1]. However, little is known about the environmental controls and phenology of sexual events in micro-eukaryotes[2–4], including diatoms, an extraordinarily diverse and ubiquitous group of unicellular algae for which sexual reproduction has been extensively studied using laboratory experiments[5,6]. During the vegetative phase of their diplontic life cycle, mitotic divisions are accompanied by a gradual decrease in the average cell size of diatom populations due to constraints imposed by their rigid silica exoskeleton[5]. When they reach a critical sexual size threshold, cells become capable of producing haploid gametes that, upon fusion, form an expanding auxospore that restores the maximum cell size. Diatom reproductive strategies broadly fall into two categories: oogamous centric diatoms, producing eggs and flagellated sperm cells, are typically homothallic, while (an)isogamous pennate

diatoms, producing non-flagellated gametes, are predominantly heterothallic (Fig. 1a).

As a result of their remarkable cell size reduction-restitution life cycle, diatom populations are expected to regularly undergo sexual reproduction to prevent critical cell miniaturization. Yet, direct field observations of this phenomenon have been sporadic[7–13] because of the short duration—from hours to days—that sexual cells persist in the environment[6], coupled with the challenge of identifying them through conventional light microscopy[14]. However, analysis of cell size distribution data[15–18], population genetic studies[19–23] and meta-transcriptomic datasets[24] have provided indirect evidence for the occurrence of sexual events.

In this study, we use an integrative approach to develop a set of sex marker genes which can detect diatom sexual reproduction within natural populations. Initially, we combine time-resolved tran-scriptomic data from controlled RNA-seq experiments to identify conserved and specific marker genes associated with diatom sexual reproduction. Subsequently, we assess their applicability and sensi-tivity through two metatranscriptomic studies. First, we investigate the potential for sexual reproduction in a microcosm experiment where we expose an estuarine diatom bloom community to sharp salinity increases. Next, we determine the occurrence of sexual events in the global ocean using metatranscriptomic data collected during the *Tara* Oceans expedition, linking them to individual Metagenome-Assembled Genomes (MAGs). The prevalence of sexual events is sup-ported by automated imaging and further validated through meta-barcoding and environmental data analysis. Our results suggest that the regulation of diatom sexual reproduction involves a complex interplay among local abundance, endogenous controls and environ-mental triggers, highlighting the need for further exploration of this key process at the base of diatom adaptation to rapidly changing aquatic environments.

## Results

### Conserved transcriptional activity during sexual reproduction

To identify conserved marker genes associated with sexual reproduc-tion, we analyzed RNA-seq data from four diatom species representing different lifestyles and reproductive strategies: the centric diatom *Ske-letonema marinoi*, the planktonic pennate diatom *Pseudo-nitzschia multistriata*, and the benthic pennate species *Seminavis robusta* and *Cylindrotheca closterium* (Fig. 1a, Tables S1, S2). To facilitate inter-species comparisons across experimental time series with varying sampling frequencies and durations of sexual reproduction, each time point was classified based on the predominant sexual stage: sex pheromone sig-naling, gametangia, gametes/zygotes, and auxospores. Multi-dimensional scaling of gene expression showed profound changes throughout the mating process (Fig. S1a, b). A total number of genes ranging from 1746 (*S. marinoi*) to 8557 (*S. robusta*) was differentially expressed (DE) during sexual reproduction compared to vegetative conditions (Fig. 1b). Upregulated genes included well-known diatom sex genes such as SIGs (Sex Induced Genes) and B-type cyclins[25,26]. *S. marinoi* exhibited a smaller number of downregulated genes, likely due to the absence of a pheromone-induced growth arrest[25], which could lead to a general population-wide downregulation of vegetative genes, even in cells not actively engaging in sex.

Using gene families that define homology between species, we next assessed the conservation of gene expression levels across dif-ferent diatom species. On average, $221 \pm 144$ (st. dev.) gene families encoded DE genes at the same sexual stage across two different spe-cies (Fig. 1c). To assess the degree and direction (up- or down-regulation) of this conserved response to sex, we evaluated the Pearson correlation of fold changes among shared gene families (Fig. 1c). Within the same species, correlations between adjacent time points were typically very high (ranging from 0.5 to 1) and always significantly different from 0 (Fig. S1c). Similarly, corresponding time

points across species showed positive correlations (ranging from 0 to 0.5), even when comparing pennate and centric diatoms, which diverged circa 150 million years ago and employ completely different mating strategies[27]. Hence, a core transcriptional program is activated during sexual reproduction of both pennate and centric diatoms. Many genes involved in DNA replication and meiosis, as well as con-served signaling modules such as members of the Deltex E3 ubiquitin-protein ligase family, were upregulated in all four species during sexual reproduction (Figs. S2a, S3, S4). Conversely, shared downregulated gene families pointed towards an inhibition of primary metabolism and photosynthesis (Fig. S2b), consistent with previous reports[28].

### Identification of eight sex-specific transcriptomic markers

The identification of a shared transcriptional response across the four species prompted us to systematically search for conserved marker genes associated with sexual reproduction (Fig. 2). First, we selected diatom-specific gene families that were expressed in at least three out of four species and were consistently upregulated during sexual reproduction. This approach allowed us to identify markers specific to pennate diatoms, as well as those shared between centric and pennate diatoms. To ensure reliability, we prioritized markers that were sex-specific, i.e., showed negligible expression under non-sexual condi-tions. Therefore, gene families were ranked by their average fold change, a measure of the difference in expression between sexual and control conditions. The top-10 ranked families (M1-M10) exhibited more than 10-fold average upregulation during sexual reproduction, with the top-5 exceeding 30-fold upregulation (Fig. 3a). To verify marker specificity, we examined the *S. robusta* expression atlas, a dataset of 119 RNA-seq samples covering diverse conditions such as toxin exposure, nutrient depletion, environmental stress, diel cycles and bacterial interactions[29,30]. The top-4 ranked marker families (M1-M4) showed negligible expression across all 42 non-sexual conditions (Fig. 3b, Fig. S5), ensuring their suitability as robust markers. On the other hand, the other families showed similar expression levels under some non-sexual and sexual conditions (Fig. 3b). Hence, to minimize false positives when screening natural samples, only M1-M4 were selected as markers. Besides RNA-sequencing, we validated these markers using reverse transcription quantitative PCR (RT-qPCR) in *C. closterium* sexual crosses (Note S1, Fig. S6, Table S3).

In terms of their function, the M1 family encodes tubby-like pro-teins, suggesting a role in gene regulation, while M2 contains trans-membrane lectin-like proteins. The M3 genes encode *SIG7*, a presumed distant homolog of the eukaryotic meiosis gene *HOP2*[25,31]. Finally, M4 consists of proteins with unknown functions containing 6-10 trans-membrane domains. While the conserved expression of M3 genes in gametangia aligns with their role during meiosis, M1, M2 and M4 were expressed at different sexual stages across different species, under-scoring the need for further work to elucidate their specific functional roles (Fig. 3c).

Among the four markers identified, only M3 is present in both centric and pennate diatoms (Fig. 3d). M1, M2 and M4 are restricted to raphid pennate diatoms and are absent in centric and araphid pennate diatoms, an ancestral subgroup of pennate diatoms from which the raphid clade diverged approximately 120 million years ago[27] (Fig. 3d). To identify additional markers for centric diatoms, we focused on genes associated with flagellum formation and function, as flagella are characteristic of sperm cells in centric diatoms. We gathered pre-viously described proteins from the diatom flagellar body and the mastigonemes (flagellar hairs) and confirmed their expression speci-ficity during gametogenesis in *S. marinoi*[25,32] (Fig. 2). Four genes were finally selected as flagella markers: *SIG1* ("Sex Induced Gene 1"), *DNAH9/11/17* and *DNAH5/8* ("Dynein Axonemal Heavy Chain") and *DRC4* ("Dynein Regulatory Complex") (Table S6).

Collectively, we identified a set of eight sex-specific marker families: one conserved across all major diatom clades, three specific

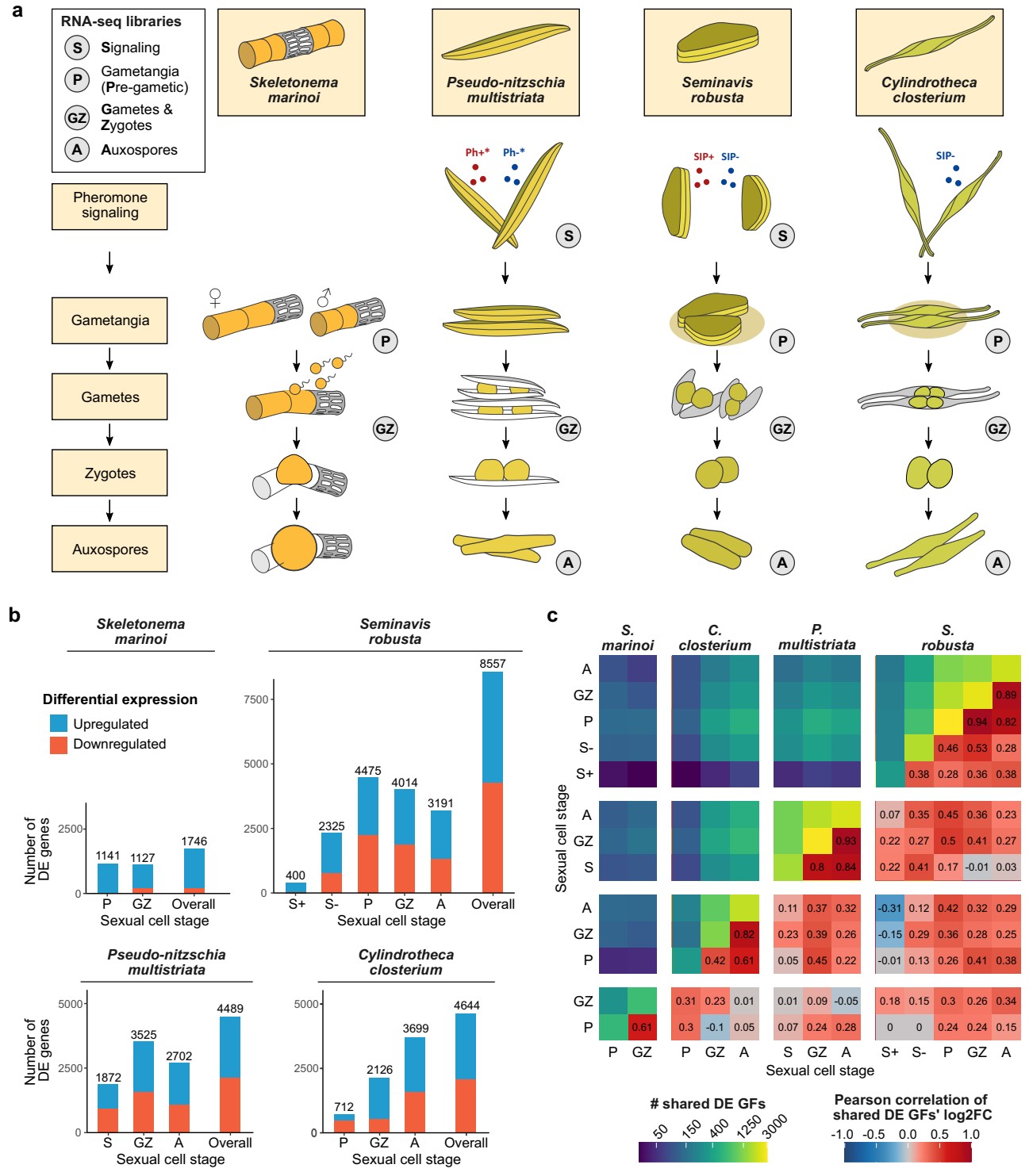

**Fig. 1 | Sexual stages, differential expression analysis and data integration of four diatom species. a** Drawings of the major cell stages during sexual reproduction in one centric and three pennate species. Stages for which RNA-sequencing data are included in our comparative analysis are indicated by encircled letters: (S) pheromone Signaling, (P) gametangia - Pre-gametic, (GZ) Gametes and Zygotes, (A) Auxospores. SIP sex-inducing pheromone, Ph-* and Ph+*: hypothetical sex pheromones of *P. multistriata* **b** Stacked bar plots showing the number of differentially expressed (DE) genes on a 5% false discovery rate level for each sexual cell stage of each species. "Overall" includes genes DE in at least one cell stage. Bars are colored to show the proportion of up- and down-regulated genes during sexual reproduction relative to the vegetative controls. **c** Heatmap showing cross-species and cross-stage expression conservation during sex. Upper triangle: number of shared homologous gene families, whose genes are DE in both sexual stages/species. Diagonal: number of gene families with DE genes in a given stage/species. Lower triangle: Pearson correlations between the average fold changes of significantly DE genes for each homologous gene family. Correlation coefficients are shown within each tile. GFs gene families. Log2FC log2(fold change).

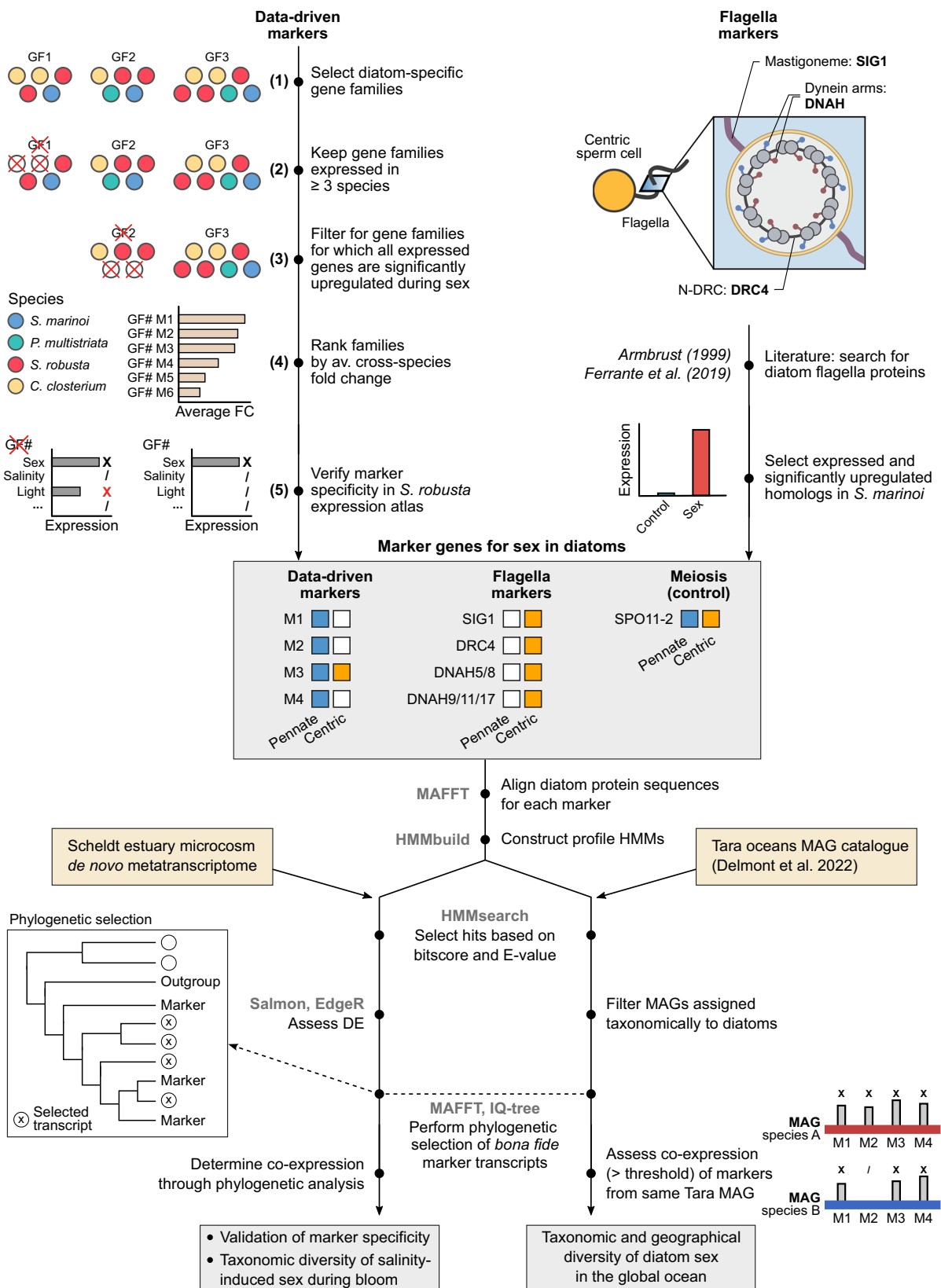

**Fig. 2 | Computational pipeline for the discovery of marker genes for sex and their detection in metatranscriptomic datasets.** This pipeline includes the five-step procedure used to identify marker gene families (GFs) in a data-driven manner, as well as the identification of flagella markers that specifically target sperm cells of centric diatoms. Subsequently, the co-expression of marker genes was assessed in the Scheldt microcosm metatranscriptome and using metatranscriptomic data linked to Metagenome-Assembled Genomes (MAGs) obtained from the *Tara* oceans expedition. Bioinformatics software used is indicated in gray. DE differential expression.

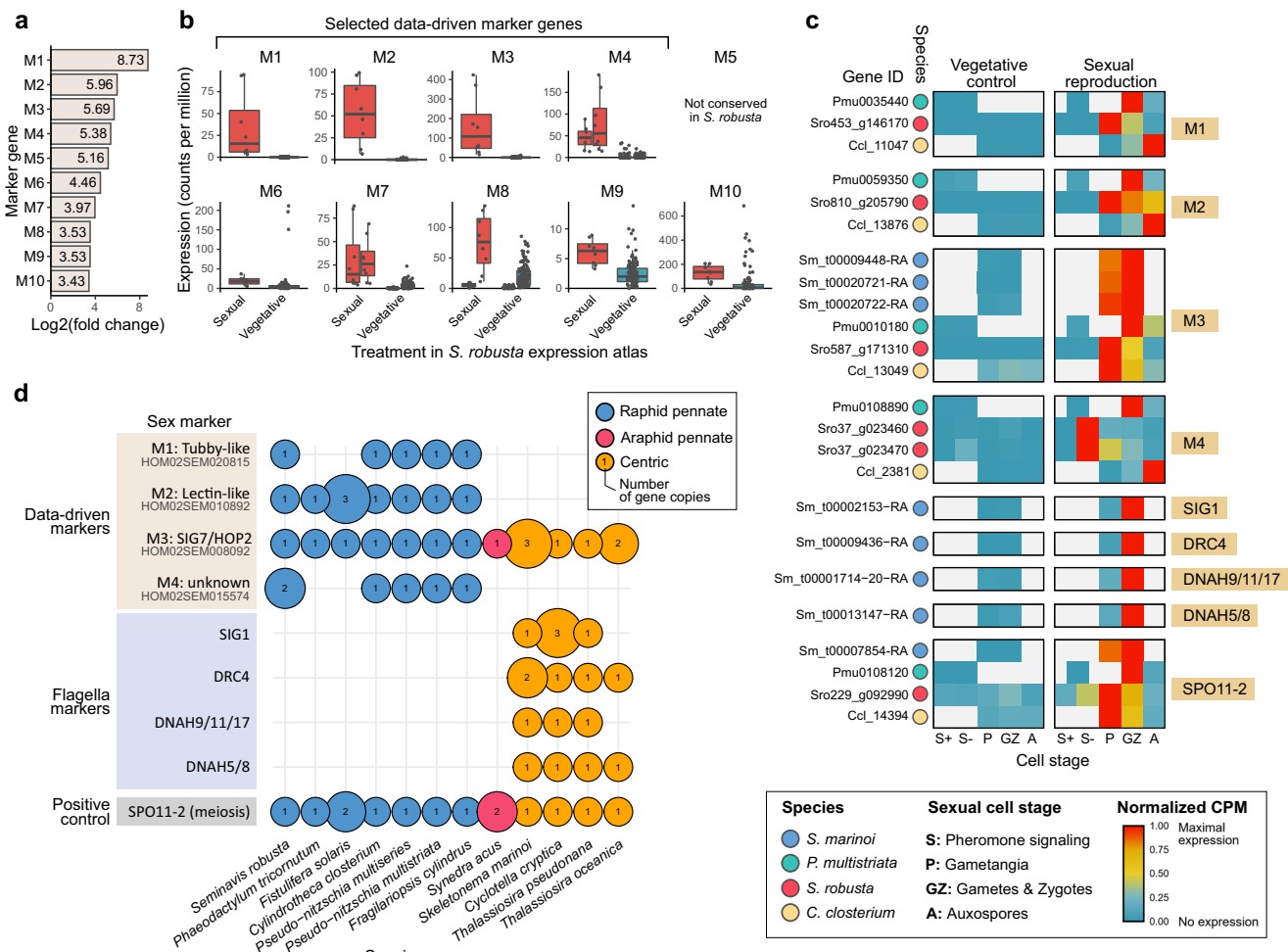

**Fig. 3 | Sex specificity, timing and species distribution of sexual reproduction markers. a** Bar plot showing the average cross-species log2 fold changes of the top-10 ranked data-driven markers during sexual reproduction. **b** Boxplots assessing expression specificity by contrasting the expression of data-driven markers during sexual reproduction ($n = 8$, pink) and other conditions ($n = 119$, blue) in the *S. robusta* expression atlas. The central line of the boxplot indicates the median, the box limits show the 25th and 75th percentiles and whiskers extend up to 1.5× the interquartile range. Individual observations are shown as dots. Markers 4, 7 and 8 contain two *S. robusta* homologs, whose expression is shown side-by-side. **c** Heatmaps showing gene expression of data-driven markers, flagella markers and the meiotic positive control gene *SPO11-2* in bulk RNA-seq datasets of sexual reproduction in four diatom species. Gene expression in normalized counts per million (CPM) of replicates was averaged for each time point. Normalized CPMs are raw CPMs that were scaled to a maximum of one per gene (row-wise). Time points are designated by a letter referring to the corresponding sexual cell stage at that time point. Empty tiles represent sexual stages that were not sampled in a given species. Expression values of the partial gene models making up *S. marinoi DNAH9/ 11/17* were summed. **d** Taxonomic distribution of sex markers in the genome assemblies of selected diatom species. The size of circles is proportional with the number of gene copies encoded in a species, while their color indicates the taxon each species belongs to.

to raphid pennates, and four flagella markers for centric diatoms (Fig. 3d). These markers are well-suited for phylogenetic analysis due to several key attributes. First, each marker family forms a discrete clade of mostly single-copy orthologs (Fig. 3d); secondly, the sequence variability within individual genes is sufficient to distinguish them at the species level. Next, we used profile hidden Markov models (HMMs), which encapsulate the sequence variation in each marker family, to assess their (co-)expression patterns in environmental metatranscriptomic datasets (Fig. 2).

**Broad sexual potential during an estuary diatom bloom**
To assess the applicability and taxonomic resolution of the sex marker genes, we experimentally induced sexual reproduction during an intense seasonal diatom bloom in the freshwater tidal part of the Scheldt estuary in Belgium, one of the last North-West European estuaries with an intact salinity gradient[33]. At the time of sampling, the bloom had almost reached its peak density (chlorophyll $a$ ~ 50 µg/L, Fig. S7) and was dominated by thalassiosiroid centric diatoms[34].

Inspired by previous laboratory studies showing that salinity can trigger sexual reproduction in certain diatom species[35,36], whole-plankton samples (salinity -0.4 ppt) were manipulated in a controlled microcosm set-up (Fig. 4a). An abrupt shift to a salinity of 10 ppt was introduced to simulate the effect of salt water incursions typical of periods of low river discharge in the estuary. Microscopic observations confirmed that this salinity shock triggered sexual reproduction in the dominant genus *Cyclotella* (Fig. 4b). In contrast to control conditions, where no sexual stages were detected during the course of the experiment, we observed spermatogenesis and fertilization of egg cells after 24 h of salinity treatment, followed by the development of auxospores after 48 h (Fig. S8–13). At both time points, 18S rDNA metabarcoding confirmed the dominance of *Cyclotella*, accounting for about 50% of the sequencing library (Fig. 4c). Phylogenetic classification of amplicon sequence variants (ASVs) revealed a single pre-dominant ASV closely related to *Cyclotella scaldensis*, which constituted 46% of all reads (Fig. 4d, e, Fig. S14). More than 20 other *C. scaldensis*-like ASVs were identified, but at much lower abundances

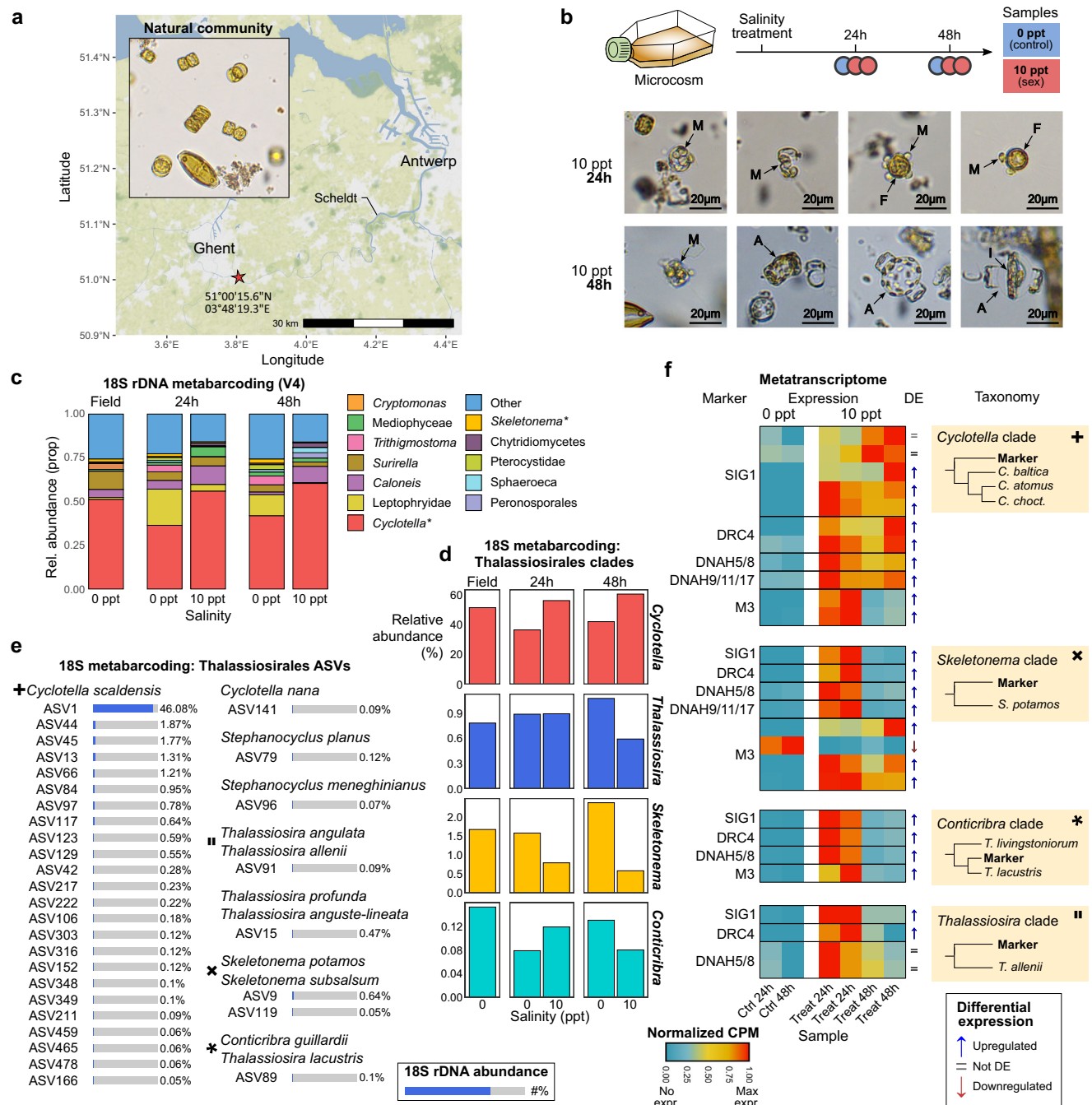

**Fig. 4 | Detection of marker genes for sexual reproduction during a microcosm metatranscriptome experiment. a** Map of the Scheldt estuary. The red star indicates the sampling location of a natural phytoplankton community for metatranscriptome sequencing. **b** Experimental design of the microcosm experiment. Microscopic images show stages of sexual reproduction of centric diatoms observed 24 h and 48 h after salinity treatment. Arrows indicate male spermatocytes and sperm cells (M), female gametes (F), auxospores (A) and an initial cell forming inside an auxospore (I). Micrographs are representative of a series of images captured during the single microcosm experiment (*n* = 1). Ppt: parts per thousand. **c** Bar plot showing the relative abundance of the 12 most abundant genera of unicellular eukaryotes detected in the microcosm experiment using 18S rDNA gene metabarcoding. Thalassiosirales species are marked by an asterisk. **d** Bar plot showing the relative 18S rDNA abundance of four Thalassiosirales genera in function of salinity treatment and time **e** Bar plots showing the average abundance (%) of 18S rDNA ASVs belonging to the Thalassiosirales. All 32 ASVs making up more than 0.1% of the library in at least one sample are shown. ASVs were classified by species or a clade of species based on phylogenetic analysis (Fig. S14). **f** Heatmaps showing the expression (normalized counts per million, CPM) and differential expression results of sex marker genes during the microcosm experiment. Heatmaps are shown for the four taxa that were co-expressing at least three markers (Fig. S18–S22). Normalized CPMs are raw CPMs that were scaled to a maximum of one per transcript (row-wise). The bold shapes connect sex markers to ASVs based on corresponding phylogenetic positions (**e**).

(<2%). In addition, other Thalassiosirales species from the genera *Conticribra*, *Skeletonema*, *Cyclotella (Stephanocyclus)* and *Thalassiosira* were present at low abundances (<1%) (Fig. 4d, e, Fig. S14).

We next performed deep metatranscriptomic sequencing (>100 million reads per replicate) to assess the co-expression of sex markers. A de novo transcriptome assembly yielded 868,920 individual transcripts (Figs. S15, S16, S17, Note S2). Among 102,859 expression-filtered transcripts belonging to four main Thalassiosirales clades (*Cyclotella*, *Thalassiosira*, *Skeletonema*, *Conticribra*), 21,308 were differentially expressed between the salinity treatment and the freshwater control conditions. For each of the sex markers, we used HMM searches to identify homologs in the de novo transcriptome. Phylogenetic analysis was then performed to retain only those transcripts that clustered within the clade of reference diatom markers (Fig. 2). In general, marker genes were significantly upregulated in response to the sex-inducing salinity treatment, demonstrating their efficacy to detect sexual reproduction (Fig. 4f). All markers showed the highest expression 24 h after salinity treatment, coinciding with flagella formation and meiosis. Phylogenetic analysis, including reference proteomes of 99 different diatom accessions, identified four distinct species that co-expressed at least three sex markers, each belonging to a different Thalassiosirales clade (Fig. 4f, Fig. S18–S22). Cross-referencing the phylogenetic position of the sex markers with the 18S barcoding data confirmed that each sexually active species corresponded to one or multiple ASVs. Interestingly, sexual reproduction was not only detected in the very abundant *Cyclotella scaldensis*, but also in rare species, such as *Thalassiosira allenii/angulata* and a *Conticribra sp.*, which represented only about 0.1% of the metabarcoding library.

## Widespread diatom sexual reproduction in the global ocean

To detect sexual reproduction in the global ocean, we examined the expression of marker genes across the 52 diatom MAGs obtained from 143 locations sampled during the *Tara* Oceans expedition (Fig. 2, Fig. S23). Despite being selected for specificity, sex markers tend to show a low level of background expression in non-sexual conditions (Fig. S5, Table S8). Therefore, we only considered marker genes to be sexually expressed when they exceeded the expression level of at least 95% of laboratory RNA-seq samples in vegetative conditions. To identify populations emitting a sexual signal in the global ocean, we focused on diatom MAGs that co-expressed sex markers in addition to the meiotic gene *SPO11-2* as a positive control. We tested the effect of requiring different degrees of co-expression (*SPO11-2*+1 marker, *SPO11-2*+2 markers, *SPO11-2*+3 markers, *SPO11-2*+4 markers) (Fig. S24). After matching the predicted sexual *Tara* stations with imaging data (see below), we decided to consider MAGs co-expressing at least two markers in addition to *SPO11-2* as indication for ongoing sexual reproduction within these populations. Using these criteria, we identified 32 and 181 cases with a co-expression signal for pennate and centric diatoms, respectively (a case referring to a MAG that shows a signal in a single sample).

In order to evaluate whether background vegetative expression of sex markers may cause false positive hits, we defined a set of five control marker families. Apart from not being upregulated during sex, control markers have similar characteristics to pennate sex markers M1-M4 and *SPO11-2*: they belong to mostly single-copy ortholog families encoded in at least three of the reference species and exhibit similar expression levels to sex markers in non-sexual conditions (Figs. S25, S26, Table S9). Repeating the co-expression analysis outlined above, we identified 7 cases where control markers showed a co-expression signal. While this is lower than the sex markers, it shows that false positives may occur under background expression levels. Hence, while co-expression of sex markers data can be an indication that sex is occurring, it should be verified with independent evidence (e.g., imaging) whenever possible.

Surprisingly, co-expression of sex markers suggested ongoing sexual activity in 9 out of 12 known diatom genera across 54 locations in the global ocean (Fig. 5a, Note S4), corresponding to 38% of all investigated stations and 46% of the examined MAGs (Table S10). MAGs exhibited distinct geographical patterns of co-expression, which did not simply reflect the overall abundance and distribution of the respective genera (Figs. S27, S28). Interestingly, for the majority (5 out of 9) of genera, we observed that multiple congeneric MAGs displayed a sexual signal in exactly the same location, suggesting co-regulation of closely related species by similar environmental parameters. *Chaetoceros*, *Fragilariopsis*, *Leptocylindrus*, *Odontella*, and *Skeletonema* showed sexual reproduction mostly in cold waters, while signals of sexual events in *Cylindrotheca* and *Pseudo-nitzschia* were most often observed in temperate and tropical regions. Finally, *Minidiscus* and *Thalassiosira* showed a more widespread distribution of sexual reproduction that ranged from polar to tropical locations (Fig. 5a).

Sexually active *Chaetoceros* and *Skeletonema* species occurred across the entire Arctic Ocean. Screening of Imaging FlowCytoBot (IFCB) data acquired during the *Tara* Oceans Polar Circle expedition confirmed the presence of vegetative cells and 4- to 8-celled spermatocytes of *Chaetoceros* at Arctic stations 163 and 188 (Fig. 5b). Here, *Chaetoceros* MAGs co-expressed sexual markers in both the pico- and nanoplankton size fractions, consistent with the diameter of the observed sexual cells. At Arctic station 188, where, in addition to *Chaetoceros*, *Skeletonema* MAGs also showed co-expression of the marker genes, we identified a single auxospore of *Fragilariopsis* in IFCB data, which was supported by a single sex marker co-expressed with *SPO11-2*. (Fig. 5a, b).

In contrast, other genera exhibited more globally widespread patterns of sexual reproduction. The cosmopolitan *Cylindrotheca*, the only genus never occurring in cold waters, mainly showed a sexual signal in temperate and subtropical areas, including the Indian Ocean, the Pacific Ocean and the Mediterranean Sea. *Pseudo-nitzschia* MAGs displayed extensive sexual activity across 10 stations in the tropical Pacific Ocean (Fig. 5a).

The observed sexual signals were then compared with diatom abundance data obtained through metabarcoding of the V4 and V9 18S rDNA regions (Fig. 6a). *Chaetoceros* exhibited a significantly higher abundance for both 18S regions in stations where it displayed sexual activity (two-sided Mann Whitney $U$-test; 18S-V4: $U(76, 661) = 28,772$, $p < 0.001$, effect size = 0.262, 95% Confidence Interval = [0.116, 0.397]. 18S-V9: $U(87, 779) = 40,029$, $p < 0.001$, effect size = 0.294, 95% Confidence Interval = [0.164, 0.415].). Furthermore, chlorophyll $a$ concentration, a proxy for bulk phytoplankton biomass, was high in the Arctic and Southern Ocean stations where *Chaetoceros* was undergoing sexual reproduction, both at the surface and Deep Chlorophyll Maximum (DCM) (Fig. 6b). While we cannot fully exclude that metatranscriptomic detection of sexual reproduction might be influenced by species abundance, we observed several stations with high *Chaetoceros* abundance that did not show any sex-specific gene expression and also observed IFCB images of vegetative *Chaetoceros* cells in samples without a sex signal, suggesting that the presence of sexual reproduction markers is not solely explained by coverage or abundance thresholds. Similarly, *Pseudo-nitzschia*, *Fragilariopsis* and *Thalassiosira* were typically abundant in locations where sexual reproduction was observed, although not significantly more than the sites without sexual activity. In contrast, the co-expression of sex marker genes by locally rare species seems to suggest that other genera were reproducing sexually at very low abundances: the median relative abundance of *Odontella*, *Minidiscus*, *Skeletonema* and *Leptocylindrus* was consistently below 1% in stations where these genera underwent sexual reproduction (Fig. 6a). Finally, sexual reproduction events were detected at various times of the year (Fig. 6b), although no clear pattern emerged when considering the prevailing environmental conditions at these stations (Fig. S29).

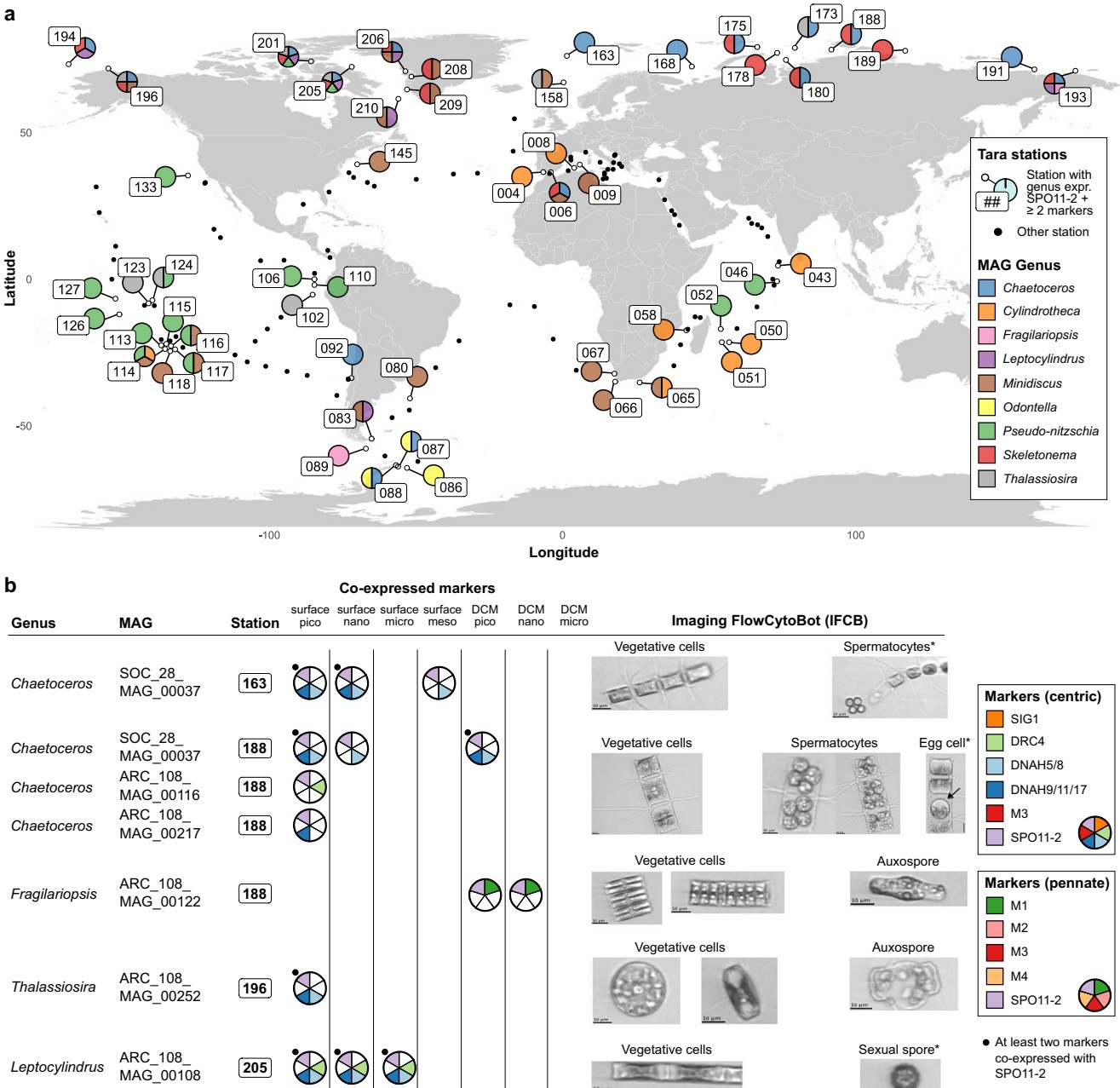

**Fig. 5 | Biogeography, taxonomy and microscopic evidence for sexual reproduction of diatoms in the global ocean. a** World map in which pie-charts indicate *Tara* stations where one or more Metagenome-Assembled Genomes (MAGs) co-express the positive control gene *SPO11-2* and at least two sex markers. Colors indicate the genus-level assignment of MAGs. Black dots indicate the other *Tara* Oceans sampling stations (total *N* = 143). **b** Microscopic evidence of sexual cell stages for the genera co-expressing sex markers in specific stations, obtained from Imaging FlowCytoBot (IFCB) data collected during the *Tara* Oceans Polar Circle expedition (https://ecotaxa.obs-vlfr.fr/). For each MAG at the given station, pie charts show the markers co-expressed with *SPO11-2* at each depth (DCM Deep Chlorophyll Maximum) and size fraction (pico: 0.8–5 µm, nano: 3–20 µm/5–20 µm, micro: 20–180 µm, meso: 180–2000 µm). Cases where at least two markers are co-expressed with *SPO11-2* are indicated with a black dot. Identifications of sexual stages with a star (*) are unsure.

## Discussion

Diatom life cycles are one of the most enigmatic aspects of phytoplankton biology. While some species have evolved mechanisms to bypass size reduction, most rely on sex to restore cell size[5], a process difficult to observe in natural environments. Understanding these reproductive strategies not only advances our knowledge of diatom biology, but also addresses one of the most fundamental questions in eukaryotic biology: why sex persists despite its apparent costs. Studying these complex processes has long been challenging, but advances in metatranscriptomics now provide a powerful tool to explore the dynamics of micro-eukaryotic communities, shedding light on species' metabolic activity[37–40], and offering novel opportunities to explore poorly understood yet critical life cycle transitions[41,42]. In this study, we integrated multiple gene expression datasets from controlled laboratory experiments and applied a rigorous computational pipeline to identify a set of eight conserved markers for sexual reproduction in diatoms. These markers are well-suited for phylogenetic analysis, allowing gene co-expression to serve as a robust and taxon-specific proxy for sexual activity.

Field observations, demographic analyses and simulation studies have suggested that sexual size restoration in diatoms occurs

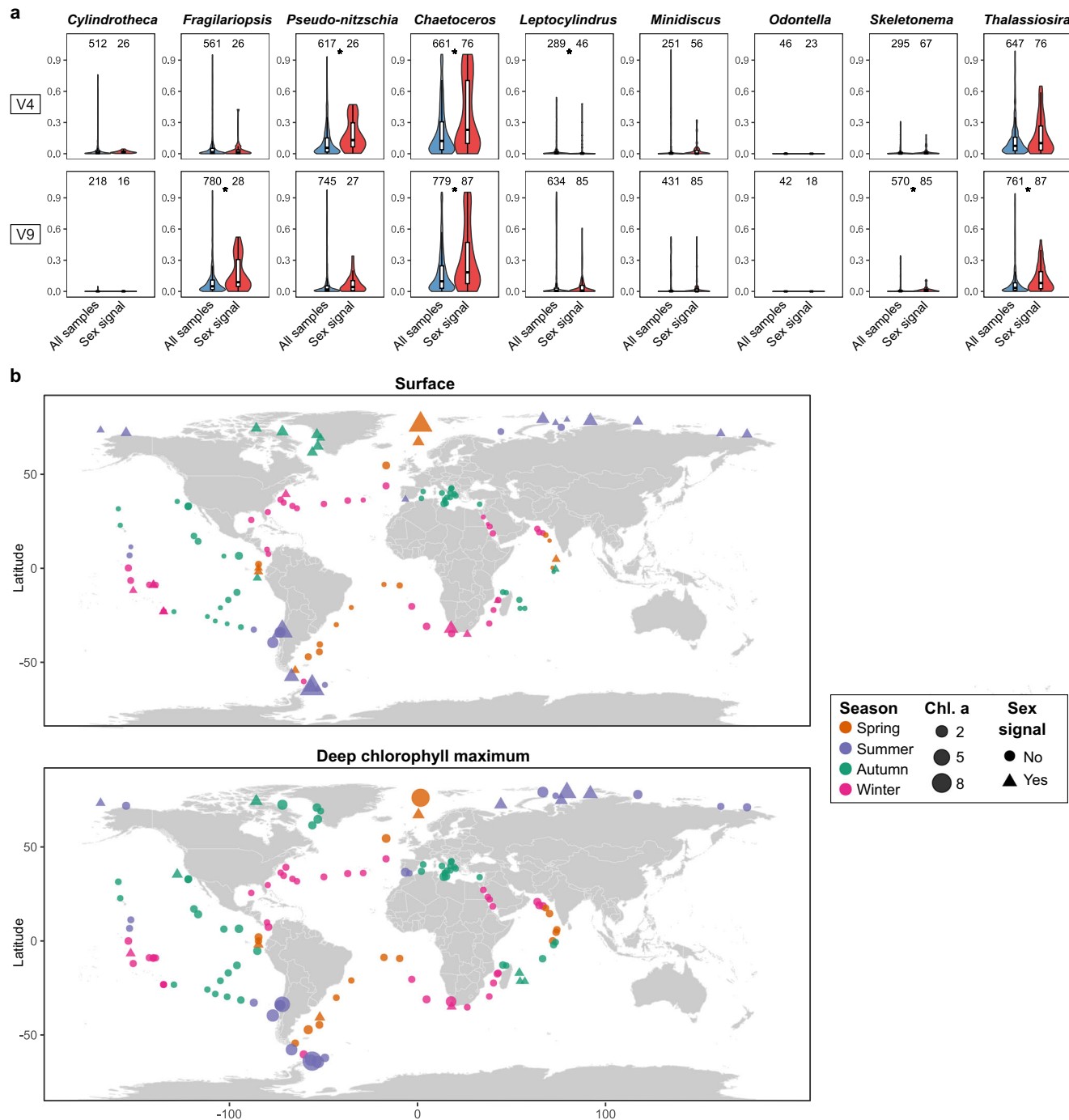

**Fig. 6 | Diatom abundance and environmental context of sexual signals during the *Tara* Oceans expedition. a** Violin plots and boxplots showing the distribution of 18S rDNA abundance (V4 and V9 regions) of different genera in the *Tara* Oceans samples relative to the total diatom abundance. For each genus, the abundance across all *Tara* samples (blue) is compared with the abundance across samples for which the genus showed a sex signal (red). Stars indicate significant differences (two-sided Mann-Whitney *U*-test with Bonferroni correction, *p* < 0.05). The number of samples is shown above each violin plot. The central line of the boxplot indicates the median, the box limits show the 25th and 75th percentiles and whiskers extend up to 1.5× the interquartile range. **b** Chlorophyll *a* concentration (mg/m³) and seasonal information, when available, across *Tara* Oceans stations in surface (top) and Deep Chlorophyll Maximum (bottom) (*N* = 137). Triangles indicate stations where we detected sexual reproduction signals, while circles indicate the other *Tara* Oceans stations. Colors indicate the season of sampling, while the size of each shape represents chlorophyll *a* concentration (mg/m³).

sporadically and in short-lived events, making it challenging to detect during routine monitoring[7–10,12,14–18,43,44]. Although the co-expression of sex markers only serves as an indirect proxy for sexual activity, the identification of a sexual signal across more than 50 *Tara* Oceans stations worldwide is notable, given the long-standing view that diatom sex is a rare event in nature. Diatom sexuality is endogenously regulated by cell size reduction, which acts as a timer to determine the interval between sexual events, thereby optimizing the balance between vegetative growth and sexual recombination[45]. The widespread occurrence of sexual reproduction suggests the co-existence of cell cohorts from different size classes, ensuring that sex can be induced whenever favorable environmental conditions arise.

*Tara* Oceans diatom genera displayed distinct geographical patterns of sexual reproduction that did not merely mirror their biogeographic distribution. Notably, sexual reproduction was most prevalent in cold-water environments, where diatoms are particularly abundant. This was evident in dominant blooming genera from the Arctic Ocean[46], such as *Chaetoceros, Fragilariopsis* and *Thalassiosira*, for which automated imaging data confirmed sexual activity. In contrast, the tropical Pacific Ocean emerged as a hotspot for sexual reproduction in the genus *Pseudo-nitzschia*. Mass sexual reproduction of *Pseudo-nitzschia* spp. has previously been observed in two different locations: the Pacific coast near Washington state (USA) and the Gulf of Naples (Italy)[10,44]. The detection of *Pseudo-nitzschia* is ecologically and economically significant, as these diatoms cause harmful algal blooms by synthesizing the neurotoxin domoic acid, with profound impacts on marine biodiversity[47]. Therefore, our set of markers provides a valuable tool for monitoring and ultimately managing *Pseudo-nitzschia* population dynamics. Interestingly, sexual reproduction patterns near the Marquesas archipelago can be linked to genus-specific preferences for iron bioavailability: *Thalassiosira* species showed a sexual reproduction signal at station 123, which is not iron-limited and is thus known for a higher abundance of *Thalassiosira* species[48].

The observed biogeographical patterns of sexual reproduction likely result from a combination of controlling factors, including spatial variations in cell size distributions, abundance and local environmental conditions. The co-occurrence of sexual reproduction among multiple MAGs of the same genus within a single sample suggests that shared environmental triggers may play a significant role in initiating sexual events. Similarly, exposure to an increased salinity induced synchronized gametogenesis in four co-occurring thalassiosiroid species from a Scheldt estuary bloom (*Cyclotella, Conticribra, Skeletonema, Thalassiosira*). A salinity shock is a known trigger for gametogenesis in euryhaline species[36,49]; in this case, it may serve as a strategy to synchronize the release of sperm and egg cells during downstream transport or in response to salt intrusions during high tide or periods of low river discharge.

Previous reports of auxosporulation have suggested a seasonal induction for most species, with sex either occurring uniquely in spring[7,8], summer[9,10], autumn[11] or even in winter[12], and overall most likely to take place in early spring and late summer in the Northern Hemisphere[13]. However, we observed that certain MAGs from genera such as *Skeletonema* and *Chaetoceros* exhibited sexual signals consistently throughout the entire sampling period in the Arctic Ocean (spring, summer, autumn). This suggests that sexual reproduction occurs continuously in a subset of the cell population below the sexual size threshold, a phenomenon known as "asynchronous sexuality" strategy[50]. This finding highlights that different species employ diverse strategies to invest in sexual reproduction. It is important to note that this observation could also be influenced by MAGs representing a consensus genome assembly of related species, each with its own ecological requirements[51].

Historical observations of sexual stages in diatoms have mostly been collected from high-density populations, raising questions about whether this reflects sampling bias or if blooming conditions indeed promote sexualization[7,9–11,44]. Comparing marker expression with amplicon sequencing data from the *Tara* Oceans dataset revealed a positive correlation between sexual events and population density in genera such as *Chaetoceros* and *Pseudo-nitzschia*. Similarly, *Cyclotella scaldensis*, a known bloom-forming species in the freshwater tidal reaches of the Scheldt estuary[52], underwent high levels of gametogenesis and auxosporulation following salt treatment, while accounting for ~50% of the entire metabarcoding library. This is further supported by laboratory experiments showing a density-dependence of sex in both *Pseudo-nitzschia* and *Cyclotella*[53,54]. Elevated population density is thought to increase encounter rates, facilitate pheromone communication, and allow post-sexual cells to escape bloom demise by sinking[55]. However,

our findings also detected sexual signals from numerous rare genera with abundances below 1% in both the Scheldt and *Tara* metatranscriptomes. While we could not microscopically confirm sexual reproduction in locally rare species, the co-expression of sex markers in both datasets suggests that sex can occur in rare species and in locations with a low phytoplankton biomass, challenging the assumption that coordinated sexual reproduction primarily occurs in high-density blooms[55]. This raises important questions about how rare species find partners in the water column, where cells are at the mercy of water circulation. Whereas motile sperm cells of centric diatoms likely rely on chemotaxis to localize conspecific egg cells, pennate diatoms largely depend on passive pairing with conspecific cells[56], for example in thin layers where the accumulation of cells increases encounter rates[53]. Alternatively, structures in the ocean could allow local accumulation of cells in association with organic matter in the form of mucus bundles, transparent gels[57] or sinking marine snow[58].

In the microcosm experiment, comparing the expression levels of markers after salinity-treatment with a control condition allowed for a robust assessment of their applicability as a proxy for sex. Similarly, a likely mass sexual reproduction event during a *Pseudo-nitzschia australis* bloom in Brittany (France) was identified by a synchronized and transient upregulation of *Pseudo-nitzschia* sex-associated genes in a metatranscriptomic time series[24]. Such multi-condition datasets provide a frame of reference to approach sex marker expression in a quantitative manner, even when using markers that are not highly specific. However, for single-sample field-based metatranscriptomic datasets, such as those from *Tara* Oceans, selecting the appropriate expression threshold to determine significant upregulation above background levels is more challenging. Here, we calculated thresholds based on the expression of sex markers in non-sexual laboratory RNA-seq experiments. It is important to note that extrapolating these quantitative data to field conditions relies on the assumption that the expression profiles of homologous genes are consistent between the field and laboratory settings. As more extensive, internally-controlled, replicated, and time-resolved metatranscriptomic datasets become available, future studies should focus on refining the quantitative aspects of sex marker expression. This will help reduce false positives and improve sensitivity in detecting rare sexual events, enhancing our ability to identify sexual reproduction in natural environments.

In conclusion, this study leveraged the precision of controlled laboratory experiments to document sexual reproduction in complex field samples. The discovery of widespread, low-abundance sexual events challenges the view that diatom sex is solely triggered by species density or bloom conditions, while geographical hotspots for individual genera suggest environmental or cell-size control of sexuality. The developed set of sex markers opens unprecedented opportunities to explore these and other aspects of the diatom life cycle. Importantly, our findings demonstrate that sexual reproduction is pervasive in the global ocean, contributing to cell size restoration and species persistence. The prevalence of sexual reproduction, possibly coupled with increased levels of mitotic recombination during stress conditions[59], helps to explain the high standing genetic diversity observed in diatom populations in the open ocean[21,60]. Genetic diversity is crucial for the evolvability of diatom species, enabling them to effectively respond to environmental changes and avoid genetic bottlenecks during bloom events[21]. Therefore, our study reinforces the notion that sexual reproduction is a key driver of the remarkable adaptability and high species diversification rates observed in diatoms[27].

## Methods
### Retrieval of diatom protein sequences, homology prediction and functional annotation
Proteomes, which represent the complete set of proteins for a species, were retrieved from the PLAZA Diatoms v1.0 platform for the 10

diatom species listed there[30]. *Cylindrotheca closterium* protein sequences were sourced from Audoor et al.[26] and *S. marinoi* nucleotide sequences of primary alleles were obtained from Audoor et al.[26] and translated to protein by TRAPID 2.0[61]. De novo transcriptome sequences for the basal centric diatom *Leptocylindrus danicus* were retrieved from the decontaminated Marine Microbial Eukaryote Transcriptome Sequencing Project (MMETSP)[62–64]. Subsequently, open reading frame prediction was carried out by a Transdecoder v5.0.2 run (github.com/TransDecoder), and homology inference was performed through a Diamond v2.0.14 search against the merged SwissProt v2022-02-09 and PLAZA Diatoms 1.0 proteome in –ultra-sensitive mode[65].

We used the homologous gene families defined in PLAZA Diatoms v1.0 to identify homologs across species[30]. As *C. closterium* and *S. marinoi* are missing from the PLAZA Diatoms platform, their genes were assigned to PLAZA gene families through TRAPID 2.0 with PLAZA Diatoms as a reference database and default settings[61]. The resulting gene family prediction for 12 genomic diatom species and 16 other eukaryotic species was used to determine the taxonomic distribution of gene families. Finally, InterPro domain information was retrieved from PLAZA Diatoms for *S. robusta* and *P. multistriata*[30], and from TRAPID 2.0 for *C. closterium* and *S. marinoi*[61].

### Differential expression analysis of sexual reproduction in four diatom species

We considered transcriptomic libraries profiling sexual reproduction in four different species: the centric diatom *S. marinoi* and the pennate diatoms *P. multistriata*, *S. robusta* and *C. closterium* (Table S1)[25,26,28,30]. To allow interspecies comparisons between time series with varying duration and sampling frequency, we assigned each sample to a mating stage based on the defining sexual cell stage present in each culture: sex pheromone signaling (S), gametangia (P, pre-gametic), gametes/zygotes (GZ) and auxospores (A) (Fig. 1a). The quality of all RNA-seq libraries was verified using FastQC v0.11.2 and summarized with MultiQC v1.7[66,67]. Illumina adapters were detected in *P. multistriata* and mating type minus *S. robusta* libraries, which were removed with Trimmomatic v0.36 (ILLUMINACLIP:2:30:10:2)[68]. FastQC was repeated to confirm successful adapter removal before proceeding to read mapping. Transcript sequences for each of the four species were derived from their respective reference genome assemblies (Table S2) and indexed with Salmon v1.3.0 with a k-mer size of 31, while retaining identical transcripts (–keepDuplicates)[69]. To improve mapping accuracy, the full set of genomic contigs of each respective species was added to the indexed transcriptome as decoys. Subsequently, expression quantification was performed for all samples with Salmon v1.3.0 in selective alignment mode and correcting for biases using the flags –gcBias and –seqBias[69]. Salmon read counts were imported into R using the tximport package[70]. For *P. multistriata, S. robusta* and *C. closterium*, tximport was used to summarize transcript counts on the gene level, while *S. marinoi* reads were already mapped to gene-level transcripts. After filtering to keep only genes showing an expression level of more than one count per million (CPM) in at least three samples and trimmed mean of *M* value (TMM) normalization, negative binomial generalized linear models were estimated for each gene using EdgeR 3.30.3[71]. Differential expression (DE) tests were performed to contrast the response between sexual and control samples for each species in each cell stage. For *P. multistriata*, gene expression of crossed cultures at each stage was compared to the average response of control cultures over both mating types, while for *S. robusta* and *C. closterium*, expression in sexual cultures was compared with their time-matched vegetative control. Meanwhile, for each *S. marinoi* cell stage, salt-treated sexual samples with a cell size below the sexual size threshold were compared with corresponding salt-treated cultures of large cell size (Table S1). For each contrast, a likelihood ratio test (LRT) was performed relative to an absolute log2 fold change threshold of

one with the glmTreat function in EdgeR. We adopted a stage-wise testing procedure as introduced in the StageR package for R to determine an "overall" set of genes at the species level, which are DE in at least one sexual cell stage, while controlling the overall false discovery rate (FDR) at 5%[72,73]. To this end, *p* values for individual contrasts were aggregated on the gene level using the Sidak method from the Aggregation package[74]. FDR-adjusted *p* values of genes that passed the screening stage were further adjusted using the stepwise Holm procedure.

### Data-driven discovery of conserved markers for sexual reproduction in diatoms

Homologous gene families allowed the integration of differential expression results between species to select sex-specific diatom families that can serve as markers. We first inspected the general association of gene expression across stages and species by calculating the Pearson correlation between the average log2 fold changes of genes differentially expressed in each gene family. Next, a five-step procedure was used to identify conserved marker genes for sexual reproduction. First, we restricted our search to diatom-specific gene families that were expressed in at least three out of four sexual diatom species. The resulting families were further filtered by keeping the ones for which all expressed genes were significantly upregulated during sex in all species where they occurred. The remaining families were ranked by the average log2 fold changes observed in the cell stages where they were significantly differentially expressed. In a final step, the sex-specific expression of the top 10 ranked gene families was verified in the 119 vegetative RNA-seq samples from the *S. robusta* expression atlas, spanning 42 different experimental treatment groups[29,30,75,76] (Table S6). Reads from the expression atlas were mapped, summarized at the gene level and normalized using TMM normalization, following the previously described pipeline. After inspecting gene expression levels in the expression atlas, the four top-ranked families (M1-4) with the highest mean fold change ranks were selected as specific markers of sexual reproduction in diatoms (Table S7).

The protein sequences of the M1-4 families were used as an input for profile HMM construction. Multiple sequence alignments for each marker gene were carried out using the MUSCLE algorithm[77] implemented in the software MEGAX, version 11.0.8[78], and then transformed into profile HMMs using the hmmbuild command of the HMMER package, version 3.1b2[79].

### Identification of meiotic genes and flagella markers

Based on literature, five diatom meiosis-specific genes with no known function outside meiosis were selected[80]: *SPO11-2, MER3, MND1, MSH4* and *MSH5*. The DNA repair gene *RAD51-A* was added as an additional meiotic gene due to its strong upregulation in *P. multistriata*[28] (Table S4). For each meiotic gene, we identified the corresponding homologous gene family on PLAZA Diatoms v1.0 and pinpointed the relevant clade containing diatom sequences and an *Ectocarpus siliculosus* outgroup (Table S4)[30]. Potential homologs in *L. danicus, S. marinoi* and *C. closterium* were identified using a relaxed BLASTp search (*E* < 1e-5) followed by construction of phylogenetic trees with IQ-tree (explained below) to verify correct clustering within the diatom clade.

Amino acid sequences of diatom flagellar proteins were extracted from two studies[25,32]. Flagella genes previously identified in *L. danicus*[64] were not considered because of their lack of expression conservation in *S. marinoi*[25]. After identification of the homologous gene family of the best BLAST hits in PLAZA Diatoms v1.0, we inspected their phylogenetic trees to select the diatom orthologs that belonged to the same subclade as the original flagella proteins, as well as outgroup sequences for phylogenetic analysis in related species *E. siliculosus* or *Aureococcus anophagefferens* (Table S5). Flagella proteins were named after the *Homo sapiens* homologs. For each flagella protein, homologs

in *S. marinoi* and *L. danicus* were identified using BLASTp ($E$ < 1e-5); the clustering of the top hit(s) in the correct diatom clade was verified by constructing phylogenetic protein trees with the PLAZA diatom homologs and outgroups. As flagella markers, we selected only those genes that were expressed during sex and significantly upregulated in the differential expression analysis of *S. marinoi* sex transcriptomes.

For each of the flagella and meiotic genes, profile HMMs were constructed using all identified homologs from diatoms. After multiple sequence alignment with MAFFT v7.453 in –auto mode[81], HMMs were constructed using the hmmbuild command from HMMER v3.1b2.

### Identification of control marker gene families

To assess the possibility that sex marker co-expression in the *Tara* dataset may be caused by background expression in vegetative conditions, we defined a set of five control markers: C1-C4 (similar to pennate sex markers M1-M4) and PC ("positive control", similar to *SPO11-2*) (Fig. S25). Control marker families were selected to have similar characteristics to the sex markers, including an expression level similar to that of the sex markers during vegetative, non-sexual conditions. Specifically, control gene families (1) must be encoded in the genome of at least three of the four reference species: *S. robusta*, *C. closterium*, *P. multistriata* and *S. marinoi*; (2) must consist of single-copy orthologs, allowing just a single duplication in one reference species; (3) must not show upregulation during sex in any of the four species. The average cross-species expression (in TPM) and its coefficient of variation (CV) were then compared to the average expression level and CV of each pennate sex marker (M1-M4, *SPO11-2*) in non-sexual conditions (Table S8). After removing families with poor phylogenetic resolution (e.g., HOM02SEM051941, the lowest expressed family in the dataset that encodes a suspected retroviral transposon), five control marker families (C1−C4 and PC) were selected (Table S9). Since only a few among the 2734 possible control families showed an expression level as low as the sex markers (Fig. S26), the final set of control markers had a higher expression level than the corresponding sex markers (Tables S8, S9).

### Microcosm experimental design, RNA extraction and sequencing

Phytoplankton abundance and bloom dynamics in the Scheldt estuary were monitored throughout the year 2022 by taking regular Chlorophyll *a* measurements. Samples were filtered over 25-mm diameter Whatman GF/F glass fiber filters, followed by High Performance Liquid Chromatography (HPLC) using the method developed by Van Heukelem and Thomas[82]. On 20/05/2022, during the early stages of a bloom of thalassiosiroid diatoms, a natural phytoplankton sample was collected from the freshwater zone of the tidal Scheldt estuary near the city of Ghent (51°00′15.6″N 3°48′19.3″E, map created with ggMap[83]). The sample was filtered through a plankton net with a mesh size of 10 μm to concentrate the plankton which was then distributed over large 150-mL culture flasks. Culture flasks were supplemented with nutrients making up WC medium with half-strength N, P and Si: 42.5 mg/l of NaNO3, 4.4 mg/l of K2HPO4 and 14.2 mg/l of Na2SiO3.9H2O and kept at 21 °C in a 12/12 day/night rhythm at 5 μmol photons $m^{-2} s^{-1}$. Half of the cultures were brought to a salinity of 10 ppt. To monitor the progression of sexual reproduction and identify species present in the sample, microscopic pictures were taken by a ToupTek TP105100A digital camera attached to an Axiovert 40 C inverted microscope fitted with a LD ACHROPLAN 20x objective and a 0.5x C-Mount (Carl Zeiss GmbH, Oberkochen, Germany).

After 24 h and 48 h, two salt-treated and one control samples were harvested by filtration on a Versapor filter with a pore size of 3 μm (Pall Corporation, NY, USA) and flash frozen in liquid nitrogen. Total RNA extraction was carried out using a RNeasy Plant Mini Kit (Qiagen) as detailed by Bilcke et al.[29]. RNA purity and quantity was assessed by NanoDrop and a bioAnalyzer Eukaryote Total RNA pico chip (Agilent) was run to verify RNA integrity. Unstranded library preparation was performed using the NEBNext Ultra RNA Library Prep Kit for Illumina including a polyA selection step to enrich for messenger RNA. Finally, 150 bp paired-end sequencing of reads was performed on the NovaSeq 6000 S4 platform at Novogene Ltd. (UK).

### Microcosm 18S metabarcoding

To assess the community composition throughout the microcosm experiment, we collected a natural sample at the sampling site and two replicates for each treatment for metabarcoding analysis. DNA samples were collected by vacuum filtration water until saturation, using a 0.22 μm MF-MilliporeTM filter. Immediately afterwards, the filter was flash frozen and stored at −80 °C, until further analysis. To assess the natural community of the sampling site, one sample was collected in the field at the moment of sampling.

DNA extraction was carried out with the DNeasy Powerlyzer Microbial Kit from Qiagen (Hilden, Germany). The polymerase chain reaction (PCR) amplification was targeting the ribosomal RNA gene. The 18S V4 region was amplified using the TAReuk454FWD1 (5′-CCAGCASCYGCGGTAATTCC-3′) and the TAReukREV3 (5′-ACTTTCGTTCTTGATYRA-3′) primers[84]. PCR and library preparation were performed following D'Hondt et al.[85]. Paired-end (2 × 300 bp) sequencing was performed with the Illumina MiSeq technology (Illumina, San Diego, US) by Genewiz (Leipzig, Germany).

We used the DADA2 pipeline to assign sequencing data to amplicon sequence variants (ASVs) for each sample[86]. Quality control, trimming and filtering was done using the 'filterAndTrim' function. TrimLeft removed the primers and trimRight removed 10 bp at the end of each read. Afterwards, forward and reverse reads were truncated after 250 and 220 nucleotides (truncLen). Sequences with EE higher than 2 and ambiguities (maxN) were removed. Pair ends were merged with a minimum overlap length of 12 bp, allowing for one mismatch using the function 'mergePairs'. The 18S sequences were assigned to the Protist Ribosomal Reference database ($PR^2$) version 4.14.0 (https://github.com/pr2database/pr2database/releases/tag/v4.14.0)[87]. The ASV table was further analyzed using the phyloseq package[88]. Contaminant sequences were removed following the method of Davis et al.[89]. Reads belonging to non-identified taxa at kingdom and infra-kingdom, as well as to the phylum Metazoa and the classes Embryophyceae and Streptophyta were deleted. ASVs with a relative abundance lower than 1e-5 % were omitted.

A deeper classification of the 32 Thalassiosirales ASVs that made up more than 0.1% of the library in at least one sample was performed by constructing a phylogenetic tree that combines ASV sequences with reference Thalassiosirales 18S nucleotide sequences from NCBI GenBank. Multiple sequence alignment was carried out with MAFFT v7.453 and a phylogenetic tree was constructed with IQ-tree v2.1.3[90] with 1000 ultrafast bootstrap repeats. Bar plots showing ASV abundance were added to this tree using the ggtree and ggtreeExtra packages for R[91].

### De novo assembly, taxonomic classification and differential expression analysis of Scheldt metatranscriptome

The quality of raw Illumina reads was checked with FastQC v0.11.2 and summarized with MultiQC v1.7[66,67]. Raw reads were normalized with the bbnorm function from BBMap v38.98, with settings target = 50 min = 20 minkmers = 15[92]. Reads of the six metatranscriptome samples were pooled to produce a single de novo co-assembly using rnaSPAdes 3.14.1 with a k-mer size of 49[93], resulting in a set of 868,920 transcripts with an average length of 1113 nucleotides and an N50 value of 1545 bp. Detection and translation into protein of open reading frames (ORFs) was performed with Transdecoder v5.0.2 (available at github.com/TransDecoder). Homology-based ORF detection was enabled through Diamond v2.0.14 searches in –ultra-

sensitive mode[65] against the SwissProt v2022-02-09 and PLAZA Diatoms 1.0 proteomes.

For taxonomic classification of reads and transcripts, a custom diatom-enriched Kraken2 v2.1.2 database[94] in protein mode was created combining Kraken2 RefSeq protein sequences (Archaea, Bacteria, human, fungi, plant and protozoa), all PLAZA Diatoms v1.0 proteomes[30], the decontaminated MMETSP sequences of 53 diatom species, 87 Thalassiosirales-related proteomes[95], 7 diatom proteomes from Phycocosm (phycocosm.jgi.doe.gov: Maypse1, Fracro1, Conwei1, Chaten1, Nitput1, Nithil2, Cylclo1) and 3 Pseudo-nitzschia proteomes[96]. The MMETSP transcriptomes were first translated by transDecoder as explained above for *L. danicus*[62,63]. We then used Kraken2 v2.1.2[94] to taxonomically classify the 868,920 rnaSPAdes metatranscripts, and in parallel to classify the raw reads for control (2 concatenated samples) and sexual (4 concatenated samples) conditions in paired-end mode.

Expression quantification was performed by mapping back all paired-end reads to the indexed rnaSPAdes de novo transcriptome (k-mer size 31) with Salmon v1.8.0[69] (settings: --gcBias, --seqBias, -l A). After importing read count tables with tximport, genes with a CPM > 0.1 in at least two samples were retained, followed by TMM normalization. Next, negative binomial generalized linear models were fitted and a LRT was performed in EdgeR v3.30.3 using the glmLRT function, comparing gene expression between salt-treated and control samples[71]. To correct for changes in the relative abundance between samples, we used the Kraken2 taxonomy to perform the differential expression pipeline separately for each of the four major Thalassiosirales clades: *Cyclotella* (29,910 filtered transcripts), *Thalassiosira* (31,895 filtered transcripts), *Conticribra* (17,418 filtered transcripts) and *Skeletonema* (23,636 filtered transcripts). While assembly-wide differential expression data was initially used to determine which taxa co-expressed sex markers (Figs. S18–22), the abundance-corrected data were used to further investigate the co-expression of sex markers within each taxon (Fig. 4). Raw unscaled abundance-corrected CPM expression values of sex marker hits are displayed in Fig. S30.

### Detection and phylogenetic analysis of marker genes in Scheldt metatranscriptome

To detect diatom homologs of sex marker genes in the metatranscriptome assembly, HMM searches were performed. Profile HMMs of the data-driven sex markers (M1-4) and flagella markers (Fig. 2) were queried against the translated rnaSPAdes metatranscriptome using HMMER v3.1b2 (bitscore >50 and $E < 1e-10$). Protein sequences of hits were combined with the original diatom marker proteins and selected outgroup sequences to delineate the specific clade of interest. Multiple sequence alignments were produced with MAFFT v7.453[81] and gaps were trimmed with TrimAl 1.4.1[97] with a gap threshold of 0.5 except for DNAH flagella proteins, where a gap threshold of 0.2 was used due to the large number and diversity of HMM hits. Maximum-likelihood phylogenetic trees were constructed based on trimmed alignments with IQ-tree v2.1.3[90] with 1000 ultrafast bootstrap repeats and automatic ModelFinder substitution model selection. Scheldt metatranscriptome proteins clustering within the clade of the original marker genes were selected ("phylogenetic selection", Fig. 2) for downstream assessment of their differential expression, taxonomic classification and gene expression levels.

Scheldt marker transcripts were phylogenetically situated in the Thalassiosirales using 87 proteomes of Thalassiosirales and related centric diatoms[95]. Sex marker homologs were selected from these 87 proteomes using HMM searches and phylogenetic selection as described above. These reference markers and the original marker protein sequences (homologs in 12 reference diatom genomes and potential non-diatom outgroups) were merged with the selected Scheldt hits, followed by multiple sequence alignment, trimming and phylogenetic tree construction as explained in the previous paragraph. Phylogenetic trees were midpoint rooted with the Phangorn package

for R. Trees were visualized with annotation and metadata using the ggtree and ggtreeExtra packages for R[91].

### Detection of marker genes in Tara Oceans MAGs

We next identified homologs for each of the eight marker genes and *SPO11-2* in the MAGs reconstructed from *Tara* Oceans metagenomic reads[98] (Fig. 2, Table S11). In brief, protein sequences encoded on a total of 683 MAGs were retrieved from https://www.genoscope.cns.fr/tara, out of which 52 diatom MAGs were searched using the *hmmsearch* command from HMMER v3.1b2. In the case of the gene M1, we masked the carboxy-terminal Tubby-like domain before querying the *Tara* Oceans database to avoid that the widespread and phylogenetically conserved domain resulted in false positives. The output hits of sexual marker genes in MAGs were filtered by selecting only sequences belonging to diatom MAGs, showing a HMMsearch sequence score ≥ 50 and an $E$ value lower than 1e−10.

Next, a phylogenetic selection was performed to select robust diatom homologs of each marker (Fig. 2). First, the protein sequences of the selected *Tara* hits were merged with the query diatom markers (Table S7) and, when available, homologs from *L. danicus* and *Chaetoceros tenuissimus*[99]. To improve the separation of the correct marker clade, we added diatom sequences from the mitosis-specific *SPO11-3* paralog to the alignment of the meiotic gene *SPO11-2*. Similarly, in the case of the two closely related markers that code for the dynein axonemal heavy chain (i.e., *DNAH5/8* and *DNAH9/11/17*), we always included the reference sequences of the sister marker as an outgroup.

Multiple sequence alignments were then performed using MAFFT v.7.490[81] followed by automatic removal of poorly aligned positions with trimAl v.1.4.1[97] and maximum-likelihood phylogenetic tree construction with IQ-TREE v2.1.3[90], using the same settings used for the microcosm experiment. Trees were midpoint rooted and visualized and edited in R (packages: *ggtree, treeio, ape, phylotools*) and FigTree version 1.4.3[100]. For each phylogenetic tree, all homologous *Tara* genes clustering within the original marker clade were selected, taking into account tree topology and bootstrap support values (Fig. 2).

### Determining co-expression of sex marker genes in Tara Oceans

After identifying sex marker genes on diatom MAGs, metatranscriptomic data associated with each MAG were analyzed to identify diatom sexual reproduction events at a global scale. *Tara* Oceans metatranscriptomic reads corresponding to various size fractions were mapped against the MAGs using BWA-MEM2 for high-efficiency mapping, followed by processing with Samtools for sorting and duplicate removal. MAGs expressing less than 10,000 reads or 1000 genes in a sample were filtered out. MAG-level transcripts per million (TPMs) were calculated for each gene relative to the total number of reads mapped to each MAG in each sample. By calculating MAG-level TPM values, we corrected for the relative abundance of each MAG across samples, ensuring that the expression of sex marker genes was normalized relative to the overall transcriptional activity of the same MAG. This enabled consistent comparisons across stations and provided a robust basis for applying gene-specific expression thresholds determined for single species in laboratory setting, as explained below.

In parallel, expression thresholds were calculated based on the four-species bulk RNA-seq dataset (Table S8). Salmon TPM values were imported into R using tximport[70]. For each sex marker family, the 95% TPM percentile was calculated across all genes in that family during vegetative conditions. This amounted to 6 vegetative samples per gene for *P. multistriata*, 119 vegetative samples per gene for *S. robusta* (expression atlas), 9 vegetative samples per gene for *C. closterium* and 8 vegetative samples per gene for *S. marinoi* (4 salt-treated and 4 untreated controls).

Next, the *Tara* expression dataset was filtered to retain only samples in which diatom MAGs expressed the positive control gene

*SPO11-2*. For those MAGs, the co-expression of sexual markers was defined as the case where multiple sex marker genes encoded on the same MAG had MAG-level TPM values exceeding the 95th percentile expression threshold (Table S8) and were supported by at least two reads within the same *Tara* sample (Table S12). Co-expression of control markers C1-C4 with their positive control PC was determined in the same manner (Table S9). The geographical distribution of sex signals was visualized using R with *ggplot2*[101] and *Scatterpie*[102] packages, by grouping the information at the genus level. Assembly metrics for *Tara* Oceans diatom MAGs were compared between sex and non-sex samples using the Mann-Whitney *U* test (Fig. S31, Table S13). The following metrics were analyzed: ANVIO completion, that represents the proportion of the genome that is assembled, reflecting the completeness of the MAG assembly; ANVIO redundancy, referring to the redundancy of the MAG assembly and thus to the level of duplicate or repetitive sequences within the assembled genomes; N50, i.e., a measure of assembly contiguity, defined as the length of the shortest contig at which half of the total assembly length is contained in contigs of that length or longer; num_contigs, i.e., the total number of contigs in the MAG assembly; total length, i.e., the total base pair length of the assembled genome. To account for multiple comparisons, *p* values were adjusted using *Bonferroni correction*.

### Integration of metadata associated with Tara Oceans

The distribution of detected sexual reproduction events across sampling stations was analyzed for different size groups and depths. We considered the following size classes: 0.8–5 μm (picoplankton), 3–20 μm and 5–20 μm (nanoplankton), 20–180 μm (microplankton), 180–2000 μm (mesoplankton), and the following depths: surface (SRF) and DCM (Deep Chlorophyll Maximum). Furthermore, we integrated information derived from metabarcoding data (V4 and V9 regions of the 18S rDNA) to further explore the putative relation between sexual reproduction events and species abundance (Note S3). OTUs tables for V4 and V9 regions were downloaded from https://zenodo.org/records/4043757 and https://zenodo.org/records/3768510, respectively. For each genus, a Mann-Whitney *U* test was applied to assess differences in distribution of abundances, with the Bonferroni correction to account for multiple tests. An adjusted *p* < 0.05 was considered statistically significant. Results are reported as U(m, n), p, effect size, and 95% confidence intervals. Full statistical values are reported in the Results for *Chaetoceros* (V4 and V9). Finally, in order to identify common conditions in locations with diatom sexual reproduction events, we analyzed environmental parameters, including seasonality. Chlorophyll *a* concentrations were downloaded from https://doi.org/10.1594/PANGAEA.875579. To compensate for missing physicochemical values in the *Tara* Oceans in situ data set, climatological data of the World Ocean Atlas 2018[103] were preferred.

IFCB data[104] from the *Tara* Oceans Polar Circle expedition were explored through the EcoTaxa[105] platform. All publicly available images as of 19 December 2023 were screened for the presence of vegetative cells and sexual stages (spermatogonia, oogonia and auxospores). Afterwards, observations of vegetative and sexual stages were cross-referenced with the predicted sexuality at each station of the nine diatom genera that co-expressed sexual markers (*Chaetoceros, Cylindrotheca, Fragilariopsis, Leptocylindrus, Minidiscus, Odontella, Pseudo-nitzschia, Skeletonema, Thalassiosira*). Results of this microscopic screening are available on Zenodo (see "Data Availability").

### Reporting summary

Further information on research design is available in the Nature Portfolio Reporting Summary linked to this article.

## Data availability

Bulk RNA-seq dataset of diatom sexual reproduction are available at the European Nucleotide Archive (ENA) with project accessions: PRJEB37110, PRJEB11784, PRJEB35793, PRJEB36275, PRJEB49955 and PRJEB33171. New paired-end Illumina reads of the Scheldt microcosm experiment are available at ENA study PRJEB61014, under accession codes ERS14855523-ERS14855528 (metatranscriptomics) and ERS16476494-ERS16476502 (metabarcoding). *Tara* Oceans metatranscriptomic reads corresponding to various size fractions are available at ENA with accession number PRJEB6609. Amplicon sequencing (metabarcoding) of *Tara* Oceans DNA samples corresponding to size fractions for protists are available at ENA with accession numbers PRJEB6610 and PRJEB9737. All other relevant data are available on Zenodo: https://zenodo.org/records/15638841 (https://doi.org/10.5281/zenodo.11258438), including (1) profile HMMs for all markers, (2) reference proteins for each marker for phylogenetic selection, (3) manually curated gene models, (4) differential expression and gene family tables for comparative transcriptomics, (5) transcripts and differential expression tables of the Scheldt metatranscriptome (both relative to the entire co-assembly and normalized per Thalassiosirales clade), (6) expression levels (# reads, transcripts per million) of genes encoded on diatom MAGs that express SPO11-2 in the *Tara* Oceans dataset, (7) phylogenetic trees of marker proteins and their metatranscriptome homologs in which the selected clades containing orthologous markers are indicated, (8) maps with co-expression on the MAG level in the *Tara* Oceans dataset, and (9) a table with observations of sex in the public IFCB data from EcoTaxa. Source data are provided with this paper.

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

## Acknowledgements

G.B. is a postdoctoral fellow supported by Fonds Wetenschappelijk Onderzoek (FWO, 1228423N). L.C. acknowledges MARCO-BOLO (ID: 101082021) and AtlantECO (ID: 862923) projects. C.B. was supported by a PhD fellowship funded by the Open University—SZN PhD program. N.R. is supported by a PhD fellowship funded by Fonds Wetenschappelijk Onderzoek (FWO, 11L2323N). Additionally, the research in this project benefited from two ASSEMBLE+ transnational access grants: DIAREP and CRISPI and was partially supported by FWO project G001521N, BOF/GOA project No. 01G01323, as well as infrastructure funded by EMBRC Belgium-FWO project G0H3817N. This research was also partially funded by the Gordon and Betty Moore Marine Microbial Initiative, grant number 7978. Chlorophyll *a* measurements of the Scheldt estuary were conducted within the framework of OMES (Onderzoek Milieu Effecten Sigmaplan), financed by the Flemish Administration for Waterways and Maritime Affairs. We are grateful towards Eric Pelletier for his generous help with *Tara* Oceans data, as well as Flora Vincent, Fabien Lombard, Lee Karp-Boss, Pierre-Luc Grondin and Claudie Marec for generating, annotating and their advice about IFCB imaging data. Finally, we thank Koen Sabbe for sharing his knowhow about the Scheldt estuary.

## Author contributions

G.B., L.C., R.A., D.I., M.I.F., K.V., and W.V. conceived and directed the study. Comparative transcriptomic and phylogenetic analyses to define conserved markers for sex were performed by G.B., with input from R.A. and K.V.d.B. N.R. was responsible for RT-qPCR validation of markers. Field sampling, laboratory experiments, metatranscriptome sequencing and bioinformatic analysis of the Scheldt community was carried out by G.B., L.A.M., P.C., and S.D., and metabarcoding was performed by L.A.M. The identification of markers in *Tara* MAGs and phylogenetic analysis was jointly performed by L.C., R.A., and C.B. L.C. carried out co-expression analyses of sex markers associated with MAGs, analyzed their environmental context and *Tara* metabarcoding data. G. B. screened public IFCB data for sexual stages, with support from M.M., M.I.F., W.V., L.C., and C.B. The manuscript was written by G.B., L.C., R.A., D.I., M.M., M.I.F., K.V. and W.V., and was approved of by all authors.

## Competing interests

The authors declare no competing interests.

## Additional information

[1]Department of Plant Biotechnology and Bioinformatics, Ghent University, Ghent, Belgium. [2]VIB Center for Plant Systems Biology, Ghent, Belgium. [3]Protistology and Aquatic Ecology, Department of Biology, Ghent University, Ghent, Belgium. [4]Stazione Zoologica Anton Dohrn, Naples, Italy. [5]BCCM/DCG Diatoms Collection, Department of Biology, Ghent University, Ghent, Belgium. [6]Statistics and Decision Sciences, Johnson and Johnson, Beerse, Belgium. [7]National Institute of Oceanography and Applied Geophysics, Trieste, Italy. [8]VIB Center for AI & Computational Biology, VIB, Ghent, Belgium. [9]These authors contributed equally: Gust Bilcke, Lucia Campese. [15]These authors jointly supervised this work: Maria Immacolata Ferrante, Klaas Vandepoele, Wim Vyverman. ✉e-mail: Mariella.Ferrante@szn.it; Klaas.Vandepoele@psb.ugent.be; Wim.Vyverman@UGent.be

