## [Transparent Peer review file · Nature Communications]

Conserved genetic markers reveal widespread diatom sexual reproduction in the global ocean

Corresponding Author: Dr Gust Bilcke

Version 0:

Reviewer comments:

Reviewer #1

(Remarks to the Author)
Reviewer's conclusion

The submitted work gives the impression of a very profound study, touching upon one of the key stages of the life cycle of diatoms – the stage of sexual reproduction.

As for the essence of the study, it should be noted here its versatility and complexity, which was achieved by involving specialists of different profiles. In this regard, it should be emphasized that for me, as a narrowly focused specialist, it is extremely difficult to give an exhaustive assessment of all the issues discussed, from problems of reproductive biology to items of molecular genetics and bioinformatics.

Since I am sufficiently familiar with the field of reproductive biology of diatoms, I can note that the presented work contains a number of new findings that significantly expand our understanding of the mechanisms of regulation of the sexual process and sexuality of diatoms.

A set of genes were identified that are differentially expressed during sexual reproduction compared to vegetative conditions. What is very important, a set of expressed genes related to sexual reproduction was established that is universal for centric and pennate species of diatoms. At the same time, marker genes characteristic only of centric and only of pennate were identified. These genes allow the sexual process to be registered directly in a natural population.

Using this approach, the authors were able to analyze the results of metagenomic analysis performed on large areas of the World Ocean. This allowed the authors to obtain evidence of the commonness, ubiquity, and high frequency of sexual reproduction of particular diatoms in the World Ocean. Previously, cases of sexual reproduction and auxospore formation were considered rare events.

Another important result of the work can be considered the confirmation that the identified genetic markers are well-suited for phylogenetic analysis due to several key attributes. First, each marker family forms a discrete clade of mostly single-copy orthologs; secondly, the sequence variability within individual genes is sufficient to distinguish them at the species level.

From the standpoint of understanding the species and the species concept, the identified marker genes involved in sexual reproduction are much more acceptable for establishing species boundaries than the commonly used chloroplast or ribosomal genes. The accumulation of nucleotide substitutions in the latter is not sufficient evidence of completed divergence in the genetic history of species; they merely correlate with the emerging, ongoing, or completed divergence of species.

The manuscript does not contain any formal errors; the text is grammatically and stylistically verified and does not require additional revision. The work meets the expected standards, at least in field of reproductive biology of diatoms. The authors provided enough detail in the methods for the work to be reproduced

From my point of view, the paper can be published in the current form.

Nickolai Davidovich

Reviewer #2

(Remarks to the Author)

• What are the noteworthy results?

The life cycles of phytoplankton are probably their least understood characteristics. Diatoms are among the most

ecologically and biogeochemically important phytoplankton, one of the most striking groups, and yet their life cycle has remained strikingly mysterious. The structure and process of formation of the silica frustule in most diatoms is such that, with every cell division, one of the daughter cells is slightly smaller than the parent, diminishing the average size of a clone through time. Cell size in most species studied is only restored through a sexual phase. This also means the sexual phase implies a loss of carbon and silicon from the phytoplankton to sedimenting silica valves as new ones are synthesized. It has long been pointed out that there are diatoms that manage to avoid the size reduction, and also making sexuality no longer obligate, and this means that resolving the diatom life cycle also might offer insight into one of the biggest questions of eukaryotic biology, which is why sex exists and persists in spite of obvious costs. The problem is that the sexual stages in diatom life cycles are difficult to observe, so there is very little information from natural environments.

The present study offers a major breakthrough, that has been years in the waiting. Early differential gene expression studies starting over 25 years ago, then by population genetics studies and, in the past decade, with full transcriptomic studies, have looked for molecular markers of the diatom life cycle that might allow detection of diatom sex in the field. So this study really builds on 25 years of work to propose robust sets of markers based on a synthesis of careful lab studies with new models, and, for the first time, uses mesocosm and direct sampling of the environment to look for diatom sex across essentially the entire ocean.

Using a combination of the new molecular markers in the Tara dataset, combined with imaging cytometry, they clearly detect events of mass diatom sex in two arctic ocean stations. This confirms the sparse and chance observations from traditional microscopy based methods that mass sex events do occur and that they can involve many very similar species simultaneously (itself surprising). Demonstrating a possible toolset for sex detection is by itself an impressive achievement that the diatom community has been waiting for a long time for, and this alone is worthy of publication at a high level because it means we might finally have what we need to effectively study how diatoms really work in the ocean.

The final result, which is potentially most surprising, is the fact that they report detection of sexual markers in a large portion of stations, suggesting it is not that rare. In fact diatom sex occurs not just in very brief pulses, but also is occurring continuously at a low level as the authors propose, that would be a very important finding both for diatom biology and, even more fundamentally, for biological oceanography.

- Will the work be of significance to the field and related fields? How does it compare to the established literature? If the work is not original, please provide relevant references.

As discussed above, I consider the work significant. It might be of great significance. The reason it might be significant beyond phytoplankton biology and oceanography is, for diatom sex to work, especially in raphid pennate diatoms which (at least in the lab) require opposite mating types to pair, the ocean must have an unseen structure. Could this structure perhaps be the mucus parachutes identified in the latest Science paper by the group of Manu Prakash (Chajwa et al., Science 386, ead15767 (2024). DOI: 10.1126/science.ad15767)? Or something else?

It also could have importance for understanding the evolutionary maintenance of sex in eukaryotes.

- Does the work support the conclusions and claims, or is additional evidence needed?

Most of the work is very rigorous and well supported. I do have one major concern, which is about the interpretation of sex markers detected in metatranscriptomes when there are no images (e.g., by IFCB) to confirm this. The authors interpret this result as supporting possibility that sex is occurring at a low level most of the time. However, their own transcriptomic data in the lab (Fig. 3 and supplementary figures S4 and S5) as well as previous transcriptome studies both in diatoms as well as many other eukaryotes, show that transcripts of even the most sex-specific genes usually can be detected at low levels from vegetative cells. That implies that, even if only vegetative cells are present, reads from such genes would eventually be detected in the metatranscriptome is deep enough.

To try to rule this out, the authors arbitrarily considered detection only when the meiotic recombination subunit spo11-2 as well as at least two other sexual phase markers were simultaneously detected. This would be expected to be a very robust strategy. However, again, if the metatranscriptomes are deep enough, eventually one might expect at least some cases of detection of reads from spo11-2 and at least two other genes even when there are only vegetative cells present. I could not tell from the present manuscript if they were able to assess this possibility.

- Are there any flaws in the data analysis, interpretation and conclusions? Do these prohibit publication or require revision?

Definitely the issue raised above is quite complex when using metatranscriptome and MAG data, with great variability in mapping, and the text implies that the authors have put in an important effort to trying to resolve the problem. However, it is not clear the size and estimated completeness of MAGs, nor do they report how much non-sexual-specific transcripts map to the MAGs. Reporting such information more clearly could help.

I think there might at least two possible strategies that might help to make their case more convincing.

In one approach, they might select genes that are expressed at similarly low levels in vegetative cells, but that are not

upregulated in the sexual phases. How often are three of such genes from the same organism (MAG) detected at the same time? Clearly such an approach in itself could have weaknesses. It may be that the conditions under which the genes selected are expressed are found frequently in the environment but not in the lab. But, if enough low-expressed vegetative genes are selected, one could use a type of bootstrapping by testing all possible three-gene combinations of a much larger set, assuming that it is unlikely that all similarly low-expressed genes in the lab are stimulated by the same environmental trigger in nature.

Another possibility might be to conduct simulations with data using the sexual markers they have selected and the lab transcriptomic data they have. Perhaps it would be possible to estimate (by repeated simulations; again, perhaps a type of bootstrapping), how likely it would be to detect coexpression of the sexual phase genes in the metatranscriptomes from vegetative cells, assuming the species contributing to the metatranscriptoms have similar enough vegetative cell transcriptomic profiles to the representatives in the lab.

Arguably both approaches suffer that they might not be strictly quantitative because of the need to extrapolate expression data from four species in the lab to the thousands of species in the field. Nevertheless, it still would be very useful. Perhaps the vegetative expression levels in the lab are higher, perhaps lower than other species in nature, but it would at least offer an approximate estimation of the likelihood of false positives (are they likely, unlikely, or very unlikely?).

- Is the methodology sound? Does the work meet the expected standards in your field?

I find most of the work quite rigorous. Some appears to be based on synthesizing several previously published lab results, and is quite rigorous. The mesocosm approach is quite strong also. The results from the arctic stations, complemented by IFCB, is quite convincing, although I have some minor comments below that might help.

- Is there enough detail provided in the methods for the work to be reproduced?

Yes. But I do suggest providing more details on the MAGs and metatranscriptomic data in this paper, as mentioned above.

In summary, I think this work should be tracked for eventual publication. I think they can probably do some important revision to address my one principal concern. As I point out, they may be able to make the case qualitatively convincing by offering some more details, but I do hope they can do one of the

More detailed comments, suggestions, and corrections follow:

Fig. 1. It's necessary to go to Fig. S1 to understand abbreviations. Could be nice to have in the main Fig. 1 or the text to make it more fluid. S: pheromone signaling, P: gametangia, pre-gametic, GZ: gametes and zygotes, A: auxospores)

Lines 84-86: "S. marinoi exhibited a smaller number of downregulated genes, which is likely the result of the lack of a sex-induced growth arrest in this species." That could be an explanation, but it could also be just that only a portion of the population is induced, so upregulation is easy to detect than downregulation. That is, if 50% of the population is induced and 50% isn't, than in 50% of the cells there will be genes which are completely unexpressed in the vegetative stage to maximal expression, and in those cells there will be genes that go from maximal expression to essentially 0. But at the population level that would be an average change of 0 to 50% max expression for the most upregulated genes but only a change from 100% max expression to 50% max expression for the most downregulated genes. There might also be differences between centrals and pennates in how the regulation works.

Paragraph starting on line 181 refers to Fig. 3, when I think they mean Fig. 4.

Lines 191-193: "High levels of spermatogenesis and fertilization of egg cells occurred... followed by the development of numerous auxospores after 48h" What are "high" levels? Is there any way to quantify this? If not, it would be perhaps better to use a phrase that is more clearly qualitative or semi-quantitative, such as that stages were easily detected in the treatment but not the control. Would be good at least in supplementary figures to have qualitative images from treatment vs control.

Fig. 4F and associated text. F) boxplots comparing the expression levels during sexual reproduction of markers belonging to each of the four taxa, calculated as the mean CPM after 10 ppt salinity treatment." It's not so clear that if the combined reads of all markers is the best way. That is, the transcripts per gene per sexual cell are likely not the same for all genes (I would expect a cell would might need to synthesize less meiotic recombination proteins than flagellar components, for example, to get the job done). Is there a way to make some normalized index that could take into account info from the lab transcriptomic data? It might be a way to make it possible to evaluate more precisely the contributions of different species to the sexual metatranscriptome. This isn't strictly necessary for the purposes of this study, but it could be very relevant to see more precisely how sex could be simultaneous or not among closely related co-occurring species.

Lines 281, 320-329: A challenge, as I understand from Notes S3, is that their methodology relies only on detection of marker genes, as they did not find a way to do relative expression. Their strategy to rely on co-expression of spo11-2 with at least two other markers is very valuable, and it is quite convincing in the stations where IFCB images confirmed sexual stages could be detected from the corresponding genera.

I offer some technical suggestions above when discussing it. Whether or not these or similar suggestions can really be

implemented for the present study, I would suggest more cautious language. For instance, "In contrast, other genera were sexually active at very low abundances" (line 281) might be re-worded to acknowledge explicitly that they have only detected an indicator suggesting sexual activity. More importantly "Our findings challenge this notion by identifying a sexual signal across the overwhelming majority of Tara Oceans stations worldwide, thus suggesting that diatom sexual events occur frequently in the global ocean" might also be toned down.

Version 1:

Reviewer comments:

Reviewer #2

(Remarks to the Author)

I congratulate the authors for their work. The revised manuscript is even more rigorous and I think it represents a breakthrough both for diatom biology and for pelagic microbial oceanography more broadly.

They reached my highest expectations in responding to my suggestions for trying to use "control" genes to assess potential false positives. I expect the bioinformatic work to satisfy this suggestion was not trivial. I point out that this type of approach will be offer a robust strategy to investigators seeking to use metatranscriptomic and MAG data for detecting other types of events that are challenging to observe (e.g., biological interactions, rare use of alternative nutrient or energy sources).

I also appreciated that they responded to my suggestions by adding some phrases to speculate succinctly on the implications of their observations for how planktonic microbes might interact.

I have some very minor comments the authors can correct on their own as they see fit, without any need for a re-review on my part:

p. 9, line 49 - space missing "ofsperm"

p. 16, line 248 "Widespread diatom sexual reproduction in the global ocean To detect sexual reproduction in the" Seems like a section title is not properly formatted

p. 16, lines 253-255: "Therefore, we only considered marker genes sexually expressed above a threshold that corresponds to the expression in maximum 5% of laboratory RNA-seq samples in vegetative conditions." The sentence needs clarification. What is meant by "in maximum 5%"? Does it mean it is only upregulated in a maximum of 5% of all vegetative conditions tried? Or that it was only detected in 5% of vegetative samples?

Reviewer #3

(Remarks to the Author)

As a non-specialist in diatom life cycle, my comments mainly concern the methodology used to study the metatranscriptome datasets. In line with the comments of the first 2 reviewers, I found the work done to select diatom marker genes specific to sexual reproduction very good. I think that this gene set could be useful for a rapid detection of sexual events in natural populations. I am also quite convinced by their conclusions that sexual reproduction occurs in many different oceanic regions. My main concern is the important lack of normalisation procedure (sorry if I missed this in the Mat&Met) to draw conclusions about the quantification/prevalence of these reproductive events in diatom populations (both microcosm and Tara datasets). Details for each analysis are given below.

My main concern with the microcosm experiment is the way the differential expression was calculated. If I understand the methodology correctly, differential expression levels were calculated for all genes in the community and not for each taxa independently. Consequently, observed expression differences could be due to variations in cell number rather than gene regulation. As mentioned lines 209-210, most genes are downregulated because due to the mortality of non-diatom taxa. So, I think all diatom genes will appear upregulated (sex marker genes and the others) because their relative abundance has increased in the community. Could the authors show that sex marker genes are more upregulated than other diatom genes? I think this is necessary to conclude that the salt treatment induced the expression of sex genes.

I have seen in the revised version that the authors have added 3 vegetative genes as a control. Is it possible to add the expression of these genes in Figure 4e to show that not all diatoms genes are upregulated? Another possibility could be to use all *Cyclotella* transcripts as a baseline and see if *Cyclotella* sex gene markers represent a higher proportion of its transcripts in the salt condition (and do the same for each diatom genus).

I don't understand what is the conclusion of Figure 4f. As *Thalassiosira* is less abundant than *Cyclotella*, all *Thalassiosira* genes will have a lower read count. I do not see anything specific to the sexual gene markers here (except if I missed something in the normalization method).

Thalassiosira and *Conticribra* do not appear in the 18S barblot (Figure 4c) because of their low abundance. Is it possible to add a panel showing the variation of the 18S relative abundance of the four clades highlighted in panels e and f?

The methodology employed to find sex genes in Tara Oceans MAGs is good and I'm convinced by the expression of diatom sex genes in many oceanic regions (Figure 5). The use of imaging data is a nice complement to prove the presence of sexual events.

However, I'm not convinced by the conclusion that "there is more abundance of Chaetoceros in stations where there is a sexual activity" (lines 302-306). How can we exclude the possibility that sex genes are detected only because Chaetoceros is abundant enough to exceed the sequencing depth threshold? Are there oceanic stations with high abundance of Chaetoceros without the expression of sex genes? If so, this could be very interesting and should be presented. Otherwise, we could interpret the results as sex genes are expressed at low level everywhere and only when the species is abundant enough we can detect it within metatranscriptome data.

In the discussion (lines 368-369), to say that sexual reproduction is more prevalent in some stations, I think the expression of sexual genes needs to be normalized to the expression of all other genes for each diatom species. Same problem for *Thalassiosira* in station 123 (lines 380-382).

Is it possible to quantify the proportion of cell ongoing sexual reproduction with imaging datasets? I think this would be a good indication of a higher prevalence in a sample.

Other points :

Figure 1c : Statistical tests on Pearson correlations are missing.

Line 654 : Could you specify which metatranscriptomic samples you used for the assembly? I assume a co-assembly of the 6 samples?

Line 675-676 : Did you apply EdgeR differential expression directly to the 850,000 transcripts? I would be curious to see the MA plot or violin plot (coloured by taxa).

Figure 4e (and related SupFigures) : Could you explain how this normalisation between 0 and 1 was done? Is it a simple ratio per gene of the CPM values? I think it is also important to see the CPM counts here (or at least the average per line).

Lines 721 to 728 : These phylogenies used to select marker genes within Tara MAGs should be available somewhere to understand how the MAG proteins were selected according to the phylogenetic placement (as SupFigure or in a public repository).

Line 359-360 : "more frequently in the global ocean" is not clear for me. Do you mean "higher frequency of sexual events in natural diatom populations than previously though"? I think this has not been shown in the manuscript. Instead, you could write "sexual events occur in more oceanic regions than previously though".

Version 2:

Reviewer comments:

Reviewer #3

(Remarks to the Author)

In this revised version, the authors have adequately addressed most of my comments. In particular, they changed the methodology used to normalise the transcriptomic data, significantly improving the robustness of the results.

My main concern is the rebuttal figure (the red/blue barplot showing Chaetoceros abundance). I am surprised that the sex signal in Chaetoceros (the most widespread species) is restricted to just 6 stations in the Arctic Ocean). This contradicts the map Figure 5a (blue dots in at least 17 different stations in several oceans) and therefore the article's main message that the sex signal is widespread. Is there a mistake in the figure, or have I missed something?

Minor points :

Line 210-214 : The authors should avoid using terms such as "differentially expressed" or "downregulation" before normalisation per taxa, as they don't know whether the change in transcript count is due to gene regulation or individual abundance. The sentence should be rephrased or removed.

Line 680: "as explained above" is written, but the normalisation is detailed below.

Line 684: To clarify the text, I recommend explaining directly that the normalisation has been performed genus per genus and removing the sentence about normalising of the entire dataset, if this analysis is no longer used in the main text.

Line 300: There is no correlation test shown in Figure 6a. This sentence should be rephrased such as: "The sexual signals are more often detected in samples with high diatom abundance."

RESPONSE TO REVIEWERS' COMMENTS

Reviewer 1

Reviewer's conclusion

The submitted work gives the impression of a very profound study, touching upon one of the key stages of the life cycle of diatoms – the stage of sexual reproduction.

As for the essence of the study, it should be noted here its versatility and complexity, which was achieved by involving specialists of different profiles. In this regard, it should be emphasized that for me, as a narrowly focused specialist, it is extremely difficult to give an exhaustive assessment of all the issues discussed, from problems of reproductive biology to items of molecular genetics and bioinformatics.

Since I am sufficiently familiar with the field of reproductive biology of diatoms, I can note that the presented work contains a number of new findings that significantly expand our understanding of the mechanisms of regulation of the sexual process and sexuality of diatoms.

A set of genes were identified that are differentially expressed during sexual reproduction compared to vegetative conditions. What is very important, a set of expressed genes related to sexual reproduction was established that is universal for centric and pennate species of diatoms. At the same time, marker genes characteristic only of centric and only of pennate were identified. These genes allow the sexual process to be registered directly in a natural population.

Using this approach, the authors were able to analyze the results of metagenomic analysis performed on large areas of the World Ocean. This allowed the authors to obtain evidence of the commonness, ubiquity, and high frequency of sexual reproduction of particular diatoms in the World Ocean. Previously, cases of sexual reproduction and auxospore formation were considered rare events.

Another important result of the work can be considered the confirmation that the identified genetic markers are well-suited for phylogenetic analysis due to several key attributes. First, each marker family forms a discrete clade of mostly single-copy orthologs; secondly, the sequence variability within individual genes is sufficient to distinguish them at the species level. From the standpoint of understanding the species and the species concept, the identified marker genes involved in sexual reproduction are much more acceptable for establishing species boundaries than the commonly used chloroplast or ribosomal genes. The accumulation of nucleotide substitutions in the latter is not sufficient evidence of completed divergence in the genetic history of species; they merely correlate with the emerging, ongoing, or completed divergence of species.

The manuscript does not contain any formal errors; the text is grammatically and stylistically verified and does not require additional revision. The work meets the expected standards, at least in the field of reproductive biology of diatoms. The authors provided enough detail in the methods for the work to be reproduced

From my point of view, the paper can be published in the current form.

Nickolai Davidovich

We thank the Reviewer, dr. Davidovich, for his thoughtful review and especially for offering his thoughts on the use of sex markers for species delimitation through phylogenetic analysis. In fact, in a separate study in preparation, we are trying to identify sex-specific genes that are diverging (i.e., evolving different sequence variants) during an ongoing and incomplete speciation event in *Seminavis robusta*. In general, we expect that cell-cell recognition proteins may be the first to diverge during prezygotic isolation, but it should be noted that the diverging genes may differ from case to case, especially between different mating systems (heterothallic vs homothallic diatoms).

Reviewer 2

- What are the noteworthy results?

The life cycles of phytoplankton are probably their least understood characteristics. Diatoms are among the most ecologically and biogeochemically important phytoplankton, one of the most striking groups, and yet their life cycle has remained strikingly mysterious. The structure and process of formation of the silica frustule in most diatoms is such that, with every cell division, one of the daughter cells is slightly smaller than the parent, diminishing the average size of a clone through time. Cell size in most species studied is only restored through a sexual phase. This also means the sexual phase implies a loss of carbon and silicon from the phytoplankton to sedimenting silica valves as new ones are synthesized. It has long been pointed out that there are diatoms that manage to avoid the size reduction, and also making sexuality no longer obligate, and this means that resolving the diatom life cycle also might offer insight into one of the biggest questions of eukaryotic biology, which is why sex exists and persists in spite of obvious costs. The problem is that the sexual stages in diatom life cycles are difficult to observe, so there is very little information from natural environments.

The present study offers a major breakthrough, that has been years in the waiting. Early differential gene expression studies starting over 25 years ago, then by population genetics studies and, in the past decade, with full transcriptomic studies, have looked for molecular markers of the diatom life cycle that might allow detection of diatom sex in the field. So this study really builds on 25 years of work to propose robust sets of markers based on a synthesis of careful lab studies with new models, and, for the first time, uses mesocosm and direct sampling of the environment to look for diatom sex across essentially the entire ocean.

Using a combination of the new molecular markers in the Tara dataset, combined with imaging cytometry, they clearly detect events of mass diatom sex in two arctic ocean stations. This confirms the sparse and chance observations from traditional microscopy based methods that mass sex events do occur and that they can involve many very similar species simultaneously (itself surprising). Demonstrating a possible toolset for sex detection is by itself an impressive achievement that the diatom community has been waiting for a long time for, and this alone is worthy of publication at a high level because it means we might finally have what we need to effectively study how diatoms really work in the ocean.

The final result, which is potentially most surprising, is the fact that they report detection of sexual marks in a large portion of stations, suggesting it is not that rare. In fact diatom sex occurs not just in very brief pulses, but also is occurring continuously at a low level as the authors propose, that would be a very important finding both for diatom biology and, even more fundamentally, for biological oceanography.

We sincerely thank the Reviewer for their supportive comments. We especially appreciate the suggestions to improve our analyses, which we have implemented as described below.

- Will the work be of significance to the field and related fields? How does it compare to the established literature? If the work is not original, please provide relevant references.

As discussed above, I consider the work significant. It might be of great significance. The reason it might be significant beyond phytoplankton biology and oceanography is, for diatom sex to work, especially in raphid pennate diatoms which (at least in the lab) require opposite mating types to pair, the ocean must have an unseen structure. Could this structure perhaps be the mucus parachutes identified in the latest Science paper by the group of Manu Prakash (Chajwa et al., Science 386, eadl5767 (2024). DOI: 10.1126/science.adl5767)? Or something else?

How the “immotile” raphid pennate diatoms manage to pair up in the water column is indeed a fascinating topic, and the observed co-expression of sex markers in several planktonic pennate diatoms supports the idea that they are able to do so. Previously, a couple of mechanisms have been proposed: passive pairing due to sinking [1] or localization in thin layers to improve pairing [2]. However, mucus parachutes are another very promising possibility as a way to bring cells together and prevent them from sinking too quickly. We added this hypothesis to the paper!

[1] Font-Muñoz, J. S. et al. Collective sinking promotes selective cell pairing in planktonic pennate diatoms. *Proceedings of the National Academy of Sciences of the United States of America* 116, 15997–16002 (2019).

[2] Scalco, E., Stec, K., Iudicone, D., Ferrante, M. I. & Montresor, M. The dynamics of sexual phase in the marine diatom *Pseudo-nitzschia multistriata* (Bacillariophyceae). *Journal of Phycology* 50, 817–828 (2014).

It also could have importance for understanding the evolutionary maintenance of sex in eukaryotes.

- Does the work support the conclusions and claims, or is additional evidence needed?

Most of the work is very rigorous and well supported. I do have one major concern, which is about the interpretation of sex markers detected in metatranscriptomes when there are no images (e.g., by IFCB) to confirm this. The authors interpret this result as supporting possibility that sex is occurring at a low level most of the time. However, their own transcriptomic data in the lab (Fig. 3 and supplementary figures S4 and S5) as well as previous transcriptome studies both in diatoms as well as many other eukaryotes, show that transcripts of even the most sex-specific genes usually can be detected at low levels from vegetative cells. That implies that, even if only vegetative cells are present, reads from such genes would eventually be detected in the metatranscriptome is deep enough.

To try to rule this out, the authors arbitrarily considered detection only when the meiotic recombination subunit *spo11-2* as well as at least two other sexual phase markers were

simultaneously detected. This would be expected to be a very robust strategy. However, again, if the metatranscriptomes are deep enough, eventually one might expect at least some cases of detection of reads from *spo11-2* and at least two other genes even when there are only vegetative cells present. I could not tell from the present manuscript if they were able to assess this possibility.

This is a pertinent remark, and something that we had been considering as well. We would like to note that the Supplementary Figure S4, which the Reviewer refers to, shows the expression of “meiotic-specific” genes (as defined in [1]), not sex markers. For these meiotic genes, there is indeed a relatively high background expression in vegetative cells, possibly due to high levels of mitotic recombination in diatoms [2]. In fact, this result was the initial reason why we did not use known meiotic genes as markers for sex, but rather chose a data-driven approach to identify markers that are more specific for sex.

In response to this question, we gathered all available data from the lab RNA-seq experiments (including the *S. robusta* expression atlas) and plotted all expression for vegetative (control) and upregulated sexual (mating) conditions to get a better sense of the background expression of the proposed sex markers. These plots have been added as a new Supplementary Figure.

Rebuttal Figure: boxplots showing the expression in counts per million (CPM) of a panel of eight sex marker families in four diatom species (*Pmu*: *Pseudo-nitzschia multistriata*, *Sro*: *Seminavis robusta*, *Ccl*: *Cylindrotheca closterium* and *Sm*: *Skeletonema marinoi*). Expression is shown in vegetative (control, blue) and upregulated sexual (mating, red) conditions. Dots show individual replicates. The central line of the boxplot indicates the median, the box limits show the 25th and 75th percentiles and whiskers extend up to 1.5x the interquartile range.

While the data-driven approach identified more specific sex-related genes than the meiotic genes shown in Supplementary Figure S4, the Reviewer raises a valid point that even for these markers, there is some expression in vegetative cells (for instance, the considerable expression of marker M3 in *Cylindrotheca closterium*). To avoid false positives, we relied on the co-expression of multiple independent markers, but we agree that this may not be sufficient to fully control for false positives. It is not an easy problem to solve: due to the complexity of *Tara* metatranscriptome data and the associated MAGs, there is no obvious background model of false positives that we could calculate. However, based on the Reviewer's suggestions, we improved our bioinformatics pipeline to assess and reduce the level of false positives (see below).

Finally, in response to “the authors interpret this result as supporting the possibility that sex is occurring at a low level most of the time”, we would like to clarify that this surprising finding is not only supported by the *Tara* analyses but also by the microcosm analysis, where, at the time of sampling, at least four thalassiosiroid species (and more conspecific genotypes) could be sexualized by a salt treatment, suggesting that these populations are “ready to reproduce”, and part of the population may continuously undergo sexualization as they are transported into more saline estuarine waters. Interestingly, while our current manuscript was under review, a very thorough meta-analysis of sex in natural diatom populations was published, going all the way back to the 19th century [3], also suggesting that sex may occur more frequently than would be implied by recent mass-event observations. We now reference this review paper in the discussion.

[1] Patil, S. et al. Identification of the meiotic toolkit in diatoms and exploration of meiosis-specific SPO11 and RAD51 homologs in the sexual species *Pseudo-nitzschia multistriata* and *Seminavis robusta*. *BMC Genomics* 16, 930 (2015).

[2] Bulankova, P. et al. Mitotic recombination between homologous chromosomes drives genomic diversity in diatoms. *Current Biology* 1–12 (2021).

[3] Mann, D. G. & Edlund, M. B. The Ecology of Diatom Reproduction. in *Diatom Ecology* 59–83 (John Wiley & Sons, Ltd, 2024).

- Are there any flaws in the data analysis, interpretation and conclusions? Do these prohibit publication or require revision?

Definitely the issue raised above is quite complex when using metatranscriptome and MAG data, with great variability in mapping, and the text implies that the authors have put in an important effort to trying to resolve the problem. However, it is not clear the size and estimated completeness of MAGs, nor do they report how much non-sexual-specific transcripts map to the MAGs. Reporting such information more clearly could help.

We appreciate the reviewer’s concerns regarding the size and completeness of the MAGs. In the revised manuscript, we have addressed these points by providing a new supplementary table (Table S13) detailing key assembly metrics for each MAG—such as total length, N50, estimated completeness, and redundancy—alongside their taxonomic assignments and whether they exhibit a sexual reproduction signal. In addition, these assembly metrics were compared between sex and non-sex MAGs using the Mann-Whitney U test, coupled with Bonferroni correction to adjust p-values accounting for multiple comparisons. The results, shown in Supplementary Figure S30, were visualized using boxplots with significant annotations. The comparison of assembly metrics for *Tara* MAGs between sex and non-sex samples reveals that sex MAGs are more complete, as evidenced by a statistically significant higher Anvio’ completion in the sex MAGs

group ($p < 0.05$, Mann-Whitney U-test with Bonferroni correction). This was expected: since we require these MAGs to encode multiple sex markers, MAGs with a sexual signal are likely to be relatively complete. In contrast, no significant differences were observed for the other metrics, including Anvio' redundancy, N50, number of contigs and total length.

As a first step to control false positives due to vegetative expression, we have implemented expression thresholds. Previously, we considered a marker to be expressed if it showed any mapping, in order to avoid complications associated with expression quantification in Tara (the main complication being how to control for differences in relative abundance of a species between samples). In the revised version, we took inspiration from the Reviewer's suggestion to determine "baseline expression levels". We used the marker's expression levels in vegetative conditions from the experimental data to set expression thresholds above which a marker gene would be considered expressed in Tara. Specifically, we selected the 95% percentile of vegetative expression for each marker family as the threshold, measured in transcripts per million (TPM). In other words, a marker would only be considered sexually expressed if it occurs in less than 5% of bulk RNA-seq samples when cells are in vegetative conditions. As shown in the figure below, these thresholds are typically quite low (around 1 TPM), but, for example, Marker M3 has a higher cutoff value due to the aforementioned background expression in *Cylindrotheca closterium*.

Rebuttal Figure: Vegetative expression levels of eight sex marker gene families in laboratory studies. The blue histograms represent expression in vegetative conditions across

four different species: *Pseudo-nitzschia multistriata* ($n = 6$ vegetative samples), *Seminavis robusta* ($n = 119$ vegetative samples), *Cylindrotheca closterium* ($n = 9$ vegetative samples) and *Skeletonema marinoi* ($n = 8$ vegetative samples). The green vertical line shows the mean expression level, while the red line indicates the 95th percentile of vegetative expression. The arrowhead shows the highest recorded level of sexual expression.

In parallel, we calculated MAG-level TPM values for all *Tara* stations. Since the expression thresholds were determined for single (laboratory) species, we calculated *Tara* TPMs relative to the total expression of each individual MAG in a given sample, thereby correcting for the relative abundance of that genotype. It must be noted that this approach does not correct for differences in the completeness of MAGs, although this effect is likely limited because the MAGs co-expressing sex markers are overall highly complete (median value > 75%, Fig. S30).

After applying the quantitative expression thresholds, the number of *Tara* Stations with a sex signal decreases from 71 to 54. Notably, we still observe similar patterns of sex in the global ocean, including “rare” species that express sex markers, and we find similar correlations with abundance and seasonality. Generally, the sexual signal remained when sexual cell stages were observed in imaging flowcytobot data (e.g., massive *Chaetoceros* spermatogenesis in station 188). For *Fragilariopsis*, we now only find a single sex marker expressed with SPO11-2 at Station 188, even though we observed - admittedly only a single - an auxospore in that sample. We now discuss this event as a possible false negative.

All relevant Figures, supplementary figures, tables and Results sections have been updated to reflect the new expression threshold approach. In addition, we have included the expression thresholds as a Supplementary Table and uploaded the dataset containing the TPM expression of all genes encoded on Diatom MAGs to the public Zenodo repository.

In addition, as we agree that the SPO11-2 + 2 markers cutoff is somewhat arbitrary, we now present the results for different numbers of co-expressed markers (SPO11-2 + 1, 2, 3 and 4 markers) in a supplementary Figure. This allows for an assessment of which sexual hits are supported by more evidence, hence being more likely to be true positives, like the cases of some Arctic Ocean stations and two tropical Pacific stations, where different genera (*Chaetoceros*, *Leptocylindrus* and *Skeletonema* in the first case, and *Thalassiosira*, *Pseudo-nitzschia* and *Minidiscus* in the latter) exhibit a strong sexual signal by co-expressing a higher number of sexual markers together with SPO11-2.

I think there might at least two possible strategies that might help to make their case more convincing.

In one approach, they might select genes that are expressed at similarly low levels in vegetative cells, but that are not upregulated in the sexual phases. How often are three of such genes from the same organism (MAG) detected at the same time? Clearly such an approach in itself could

have weaknesses. It may be that the conditions under which the genes selected are expressed are found frequently in the environment but not in the lab. But, if enough low-expressed vegetative genes are selected, one could use a type of bootstrapping by testing all possible three-gene combinations of a much larger set, assuming that it is unlikely that all similarly low-expressed genes in the lab are stimulated by the same environmental trigger in nature.

Another possibility might be to conduct simulations with data using the sexual markers they have selected and the lab transcriptomic data they have. Perhaps it would be possible to estimate (by repeated simulations; again, perhaps a type of bootstrapping), how likely it would be to detect coexpression of the sexual phase genes in the metatranscriptomes from vegetative cells, assuming the species contributing to the metatranscriptoms have similar enough vegetative cell transcriptomic profiles to the representatives in the lab.

Arguably both approaches suffer that they might not be strictly quantitative because of the need to extrapolate expression data from four species in the lab to the thousands of species in the field. Nevertheless, it still would be very useful. Perhaps the vegetative expression levels in the lab are higher, perhaps lower than other species in nature, but it would at least offer an approximate estimation of the likelihood of false positives (are they likely, unlikely, or very unlikely?).

In addition to the expression thresholds explained above, we believe it is useful to consider the behavior of non-sexual vegetative genes in *Tara* as a baseline for the level of co-expression that could be expected by chance. Specifically, we now defined non-sexual “control markers” that are expressed at a similar level to the sex markers in vegetative conditions, and we assessed how often these control markers would be considered co-expressed in *Tara*, by following the same threshold rules as the sex markers. Since we found almost no gene families with such low expression level (see below) and as these families require manual curation and phylogenetic selection (which is very time-consuming and not straightforward to automate), we limited ourselves to a set of 5 control families that are as similar as possible to the pennate sex markers in all aspects (SPO11-2 + M1-M4). We chose to apply this approach only to the pennate markers and not to the centric (flagella) markers, because we only have one centric reference species and lack an expression atlas to accurately assess expression across a large number of vegetative conditions.

Apart from never being upregulated during sex, the control marker families had to be as similar as possible to the sex marker families: (A) encoded in the genome of a similar set/clade of diatoms, (B) mostly single-copy ortholog families, and (C) having similar expression levels (both average and variability) to the sex markers in vegetative conditions. To select these families, we used a data-driven approach similar to the one used for the pennate sex markers (see supplementary figure below). Note how selecting for conserved families in step 1 already removed most families showing very low expression.

Rebuttal Figure: bioinformatic pipeline for the discovery of “C1-C4” control marker families with similar characteristics as the data-driven sex markers M1-M4, and “PC” similar to the positive control gene SPO11-2. GFs: gene families.

Next, we identified families with a similar expression level and coefficient of variation to the sex markers during vegetative conditions. However, among the 2734 families, those with an expression as low as the sex markers were very rare, further supporting the specificity of our sex markers. In fact, only one family exhibited a lower average expression level than marker M1 (see figure below), and further inspection revealed it is a retroviral transposon with poor conservation and phylogenetic descent, making it unsuitable for use. As a result, we had to select control markers with a higher baseline expression than the sex markers, which may affect the results by suggesting a higher level of false positives.

Rebuttal Figure: scatterplot showing the coefficient of variation (CV) and average expression (transcripts per million, TPM, note the logarithmic scale). Each point represents a single gene family. Dots are coloured by potential control family (orange), data-driven sex markers (blue) and positive control gene SPO11-2 (red). Averages and CV were calculated including all gene family homologs that belong to any of the following four species: *Pseudo-nitzschia multistriata*, *Seminavis robusta*, *Skeletonema marinoi*, and *Cylindrotheca closterium*. Expression statistics of sex markers were calculated for vegetative conditions only (non-sexual samples), while for control families they were calculated over all conditions (non-sexual and sexual).

Ultimately, after selecting control markers that are as similar as possible to M1-M4 and SPO11-2, we ended up with five families: C1-C4 and PC (“positive control”, similar to SPO11-2):

We performed HMM searches against *Tara* MAG genes and conducted phylogenetic selection of pennate diatom orthologs to compare with the “pennate sex markers” M1-M4. We then considered how many *Tara* samples were co-expressing the positive control + 2 control markers and found that this occurred in seven cases (a case refers to a single MAG in a single sample). While this is

obviously lower than what happens with the sex markers (we recorded 32 cases of pennate diatoms co-expressing the sex markers and 181 cases of centric diatoms), it does show that false positives can occur at the natural background expression levels of sex markers.

We have added this control example to the manuscript and clarified that, although co-expression may signal that sex is occurring (e.g., in case of the IFCB imaged sites), false positives remain possible. We have also added a section in the discussion where we highlight the need for some kind of control for false positives in observational metatranscriptome datasets like Tara. As suggested by the Reviewer, we also discuss how the approaches based on expression in the lab rely on the key assumption that expression levels in natural species are similar to those in laboratory species.

- Is the methodology sound? Does the work meet the expected standards in your field?

I find most of the work quite rigorous. Some appears to be based on synthesizing several previously published lab results, and is quite rigorous. The mesocosm approach is quite strong also. The results from the arctic stations, complemented by IFCB, is quite convincing, although I have some minor comments below that might help.

- Is there enough detail provided in the methods for the work to be reproduced?

Yes. But I do suggest providing more details on the MAGs and metatranscriptomic data in this paper, as mentioned above.

In summary, I think this work should be tracked for eventual publication. I think they can probably do some important revision to address my one principal concern. As I point out, they may be able to make the case qualitatively convincing by offering some more details, but I do hope they can do one of the suggestions for estimating false positives.

Taken together, based on the Reviewer's advice, we improved the *Tara* bioinformatics pipeline in the following ways: (1) introducing data-driven expression thresholds, (2) visualizing the effect of different co-expression levels, and (3) introducing a simple set of control genes to assess the potential of false positives.

More detailed comments, suggestions, and corrections follow:

Fig. 1. It's necessary to go to Fig. S1 to understand abbreviations. Could be nice to have in the main Fig. 1 or the text to make it more fluid. S: pheromone signaling, P: gametangia, pre-gametic, GZ: gametes and zygotes, A: auxospores)

Thank you for the suggestion, we have clarified the abbreviations in the top left corner of the figure and in the caption.

Lines 84-86: “*S. marinoi* exhibited a smaller number of downregulated genes, which is likely the result of the lack of a sex-induced growth arrest in this species.” That could be an explanation, but it could also be just that only a portion of the population is induced, so upregulation is easy to detect than downregulation. That is, if 50% of the population is induced and 50% isn’t, than in 50% of the cells there will be genes which are completely unexpressed in the vegetative stage to maximal expression, and in those cells there will be genes that go from maximal expression to essentially 0. But at the population level that would be an average change of 0 to 50% max expression for the most upregulated genes but only a change from 100% max expression to 50% max expression for the most downregulated genes. There might also be differences between centrics and pennates in how the regulation works.

We agree with the Reviewer’s reasoning. In diatom culture experiments, there is always a specific part of the population that does not engage in sex (often much more than 50%). Therefore, it should indeed be easier to detect upregulated genes (whose fold change should approach infinity in the case of a perfect marker gene) than downregulated genes (whose expression can only decrease to the level of non-sexual cells in the culture). This is reflected in the volcano plots (Fig. S1), which are asymmetric for all species: the top upregulated genes are much more significant and have higher fold changes than the top downregulated ones.

This comment made us realize that we hadn’t sufficiently clarified this point in the manuscript. Specifically, the three pennate diatoms use sex pheromones to induce a growth arrest. This means that even a part of the population not actively engaging in sex can still be affected, leading to downregulation of cell cycle and photosynthetic genes across the whole population [1,2]. In contrast, no growth arrest and no sex pheromones affecting the entire population are known for centric diatoms. This likely results in a lack of downregulated genes in centrics.

We have rephrased this point as follows: “*S. marinoi* exhibited a smaller number of downregulated genes, which is likely the result of the lack of a pheromone-induced growth arrest that could cause a general population-wide downregulation of vegetative genes even in cells not actively engaging in sex.”

1. Annunziata, R. et al. Trade-off between sex and growth in diatoms: Molecular mechanisms and demographic implications. *Science Advances* 8, eabj9466 (2022).
2. Audoor, S. et al. Transcriptional chronology reveals conserved genes involved in pennate diatom sexual reproduction. *Mol Ecol* 33, e17320 (2024).

Paragraph starting on line 181 refers to Fig. 3, when I think they mean Fig. 4.

Thank you for spotting this oversight, we have corrected all references to this figure in this paragraph.

Lines 191-193: “High levels of spermatogenesis and fertilization of egg cells occurred... followed by the development of numerous auxospores after 48h” What are “high” levels? Is there any way

to quantify this? If not, it would be perhaps better to use a phrase that is more clearly qualitative or semi-quantitative, such as that stages were easily detected in the treatment but not the control. Would be good at least in supplementary figures to have qualitative images from treatment vs control.

We agree that the statement was too subjective, based on solely qualitative observations. Unlike in single-species laboratory cultures, we could not quantify the level of sexual reproduction in this community because it consisted of hard-to-distinguish *Cyclotella*/Thalassioriales species. Additionally, we could not differentiate empty frustules that had released sperm cells from dead cells. The estuary system also contained a significant amount of sediment and detritus, which made cell counting challenging.

In more qualitative terms, however, we never observed any sexual stage in the control (untreated microcosm) when screening these cultures. In contrast, after the salt treatment, a majority of *Cyclotella*-sized centric cells appears to have differentiated into either sperm cells (empty frustule), egg cells (surrounded by sperm cells), and later auxospores.

As suggested by the Reviewer, we have included original, uncropped microscopic images of untreated and salt-treated cultures taken during the microcosm experiment as new Supplementary Figures. In addition, we have revised the text as follows:

“In contrast to control conditions, where no sexual stages were detected during the course of the experiment, we observed spermatogenesis and fertilization of egg cells after 24 h of salinity treatment, followed by the development of auxospores after 48 h (**Figures S8-13**).”

Fig. 4F and associated text. F) boxplots comparing the expression levels during sexual reproduction of markers belonging to each of the four taxa, calculated as the mean CPM after 10 ppt salinity treatment.” It’s not so clear that if the combined reads of all markers is the best way. That is, the transcripts per gene per sexual cell are likely not the same for all genes (I would expect a cell would might need to synthesize less meiotic recombination proteins than flagellar components, for example, to get the job done). Is there a way to make some normalized index that could take into account info from the lab transcriptomic data? It might be a way to make it possible to evaluate more precisely the contributions of different species to the sexual metatranscriptome. This isn’t strictly necessary for the purposes of this study, but it could be very relevant to see more precisely how sex could be simultaneous or not among closely related co-occurring species.

The idea of using these markers to determine the intensity of a sexual reproduction event for each species is indeed very interesting. Specifically, the Reviewer suggests that the expression level of individual markers in the metatranscriptome, relative to their expression in laboratory transcriptomes, can be used to estimate the proportion of a population undergoing sex. For instance, if a population of 100% sexual cells expresses DRC4 at 100 CPM, then an expression

level of 20 CPM in the metatranscriptome would suggest that approximately 20% of cells are (1) sexual cells (2) of that particular species.

However, calculating such an “expression index per sexual cell” for each marker gene does not seem feasible with the current data for three reasons:

1. In lab experiments, only part of the population typically engages in sex. The level of sexual reproduction in the “reference” RNA-seq datasets we use is variable, making it difficult to determine what expression level can be expected for each gene in a population that is 100% sexual.
2. The expression of sex marker genes may depend on the specific stage of sexual reproduction. For instance, meiotic recombination genes might show a higher expression level just before gametogenesis, followed by a burst of expression in flagella genes during gametogenesis. Hence, the expected expression level of each marker gene would depend on the specific timing and mixture of sexual stages in the population in both the “reference” lab population and the metatranscriptome population.
3. Different species in the metatranscriptome community may have a different efficiency in RNA extraction or quality, making quantitative comparisons between species unreliable.

When we split the boxplots by marker to compare expression levels between species (see below, for simplicity only showing the highest expressed paralog), we do not observe the expected trend where certain markers always have higher/lower expression than the others. This is possibly due to point (2) above: the different species may be in different (sub)stages of sexual reproduction at the time of sampling.

Taken together, we believe that the current box plots reflect the reality quite well: there is a general trend where expression level follows species abundance, but there is considerable variability in the expression level between markers, which is too big to allow for any kind of quantitative prediction.

We hope that the set of markers proposed here will stimulate future research into this topic. More extensive time series of sexual reproduction in nature - whether in an induced setting, like in this paper, or by tracking the development of a bloom or sex event - would make it possible to link sex intensity to expression levels and connect expression timing with the succession of developmental stages.

Lines 281, 320-329: A challenge, as I understand from Notes S3, is that their methodology relies only on detection of marker genes, as they did not find a way to do relative expression. Their strategy to rely on co-expression of *spo11-2* with at least two other markers is very valuable, and it is quite convincing in the stations where IFCB images confirmed sexual stages could be detected from the corresponding genera.

I offer some technical suggestions above when discussing it. Whether or not these or similar suggestions can really be implemented for the present study, I would suggest more cautious language. For instance, “In contrast, other genera were sexually active at very low abundances” (line 281) might be re-worded to acknowledge explicitly that they have only detected an indicator suggesting sexual activity. More importantly “Our findings challenge this notion by identifying a sexual signal across the overwhelming majority of Tara Oceans stations worldwide, thus suggesting that diatom sexual events occur frequently in the global ocean” might also be toned down.

We have toned down these statements, and now also acknowledge that co-expression of sex markers is only an indicator of sex. Some examples below.

“In contrast, the co-expression of sex marker genes by locally rare species seems to suggest that other genera were sexually active at very low abundances: (...)”

“Although the co-expression of sex markers is only an indirect proxy for sexual activity, the identification of a sexual signal across the majority of Tara Oceans stations worldwide suggests that diatom sexual events may occur more frequently in the global ocean than previously thought.”

RESPONSE TO REVIEWERS' COMMENTS

Reviewer #2 (Remarks to the Author):

I congratulate the authors for their work. The revised manuscript is even more rigorous and I think it represents a breakthrough both for diatom biology and for pelagic microbial oceanography more broadly.

They reached my highest expectations in responding to my suggestions for trying to use "control" genes to assess potential false positives. I expect the bioinformatic work to satisfy this suggestion was not trivial. I point out that this type of approach will offer a robust strategy to investigators seeking to use metatranscriptomic and MAG data for detecting other types of events that are challenging to observe (e.g., biological interactions, rare use of alternative nutrient or energy sources).

I also appreciated that they responded to my suggestions by adding some phrases to speculate succinctly on the implications of their observations for how planktonic microbes might interact.

We thank the Reviewer for their supportive comments and for re-examining our manuscript in such detail. We are happy that our improvements to the previous version of the manuscript were satisfactory.

We have implemented the remaining improvements suggested by the Reviewer below.

I have some very minor comments the authors can correct on their own as they see fit, without any need for a re-review on my part:

p. 9, line 49 - space missing "ofsperm"

p. 16, line 248 "Widespread diatom sexual reproduction in the global ocean To detect sexual reproduction in the" Seems like a section title is not properly formatted

p. 16, lines 253-255: "Therefore, we only considered marker genes sexually expressed above a threshold that corresponds to the expression in maximum 5% of laboratory RNA-seq samples in vegetative conditions." The sentence needs clarification. What is meant by "in maximum 5%"? Does it mean it is only upregulated in a maximum of 5% of all vegetative conditions tried? Or that it was only detected in 5% of vegetative samples?

Reviewer #3 (Remarks to the Author):

As a non-specialist in diatom life cycle, my comments mainly concern the methodology used to study the metatranscriptome datasets. In line with the comments of the first 2 reviewers, I found the work done to select diatom marker genes specific to sexual reproduction very good. I think that this gene set could be useful for a rapid detection of sexual events in natural populations. I am also quite convinced by their conclusions that sexual reproduction occurs in many different oceanic regions. My main concern is the important lack of normalisation procedure (sorry if I missed this in the Mat&Met) to draw conclusions about the quantification/prevalence of these reproductive events in diatom populations (both microcosm and Tara datasets). Details for each analysis are given below.

We thank the Reviewer for critically assessing the methodology of our metatranscriptome analyses. We indeed previously did not perform normalization for the microcosm differential expression experiment, and have now implemented this in the revised manuscript. For the *Tara* dataset we did include a correction for taxon-abundance but this was not explained clearly enough in the manuscript. Below, we provide a point-by-point response to your concerns.

My main concern with the microcosm experiment is the way the differential expression was calculated. If I understand the methodology correctly, differential expression levels were calculated for all genes in the community and not for each taxa independently. Consequently, observed expression differences could be due to variations in cell number rather than gene regulation. As mentioned lines 209-210, most genes are downregulated due to the mortality of non-diatom taxa. So, I think all diatom genes will appear upregulated (sex marker genes and the others) because their relative abundance has increased in the community. **Could the authors show that sex marker genes are more upregulated than other diatom genes ?** I think this is necessary to conclude that the salt treatment induced the expression of sex genes.

It is correct that we did not perform any kind of normalization that takes into account the taxon-specific abundance shifts between treatments in the microcosm experiment. However, we reasoned that, despite some shifts in community composition (most strikingly the demise of *Leptophryidae* parasites after salt treatment), the changes in the relative abundance of *Thalassiosirales* diatoms were not sufficient to fully explain the observed upregulation of sex marker genes. As can be seen from the 18S rDNA barcoding barplots on the right, *Cyclotella* increased in relative abundance by about 40%, but some of the other *Thalassiosirales* clades even decreased after a salt treatment, a pattern inconsistent with the concurrent upregulation of sex marker genes.

While we previously already annotated raw microcosm RNA reads and transcripts with Kraken2, we needed a better taxonomic resolution to answer your questions, especially for the Thalassiosirales. Therefore, we created a new Kraken2 protein database that includes 97 new diatom proteomes, among which 87 thalassiosiroids. Apart from offering a higher taxonomic resolution, this also led to a better capture of reads: the number of unassigned reads decreased by ~10%. The Methods section and Supplementary Figures 15 and 16 were updated to reflect this change in the Kraken database.

We next applied Kraken2 to the transcripts from the microcosm *de novo* transcriptome. To verify the accuracy of the extended Kraken database, we checked the sex marker gene hits, which we previously manually annotated using phylogenetic analysis. Kraken2 correctly assigned all of these sex marker hits to the order Thalassiosirales and 26/27 transcripts were correctly placed among the four Thalassiosira clades that we use in the paper. Hence, these Kraken2 taxonomy predictions are accurate and can be used to determine whether expression trends are affected by community composition.

We first compared the differential expression results between the four most common taxa (Bacteria, Diatoms, Fungi+Animals, and plants) (see barplots). Surprisingly, diatom transcripts were less likely to be upregulated in response to salt compared to the other taxa (11.5% for diatoms vs 15.1% on average for other taxa). The majority of the sex marker hits (21/27) was among the 10% most upregulated Thalassiosirales transcripts in the dataset, further supporting that their response is not just a side-effect of a general shift in abundance.

Because changes in abundance may only affect the differential expression at a lower taxonomic level (e.g. the *Cyclotella* species that increased in abundance), we visualized the fold changes for each of the four Thalassiosirales clades separately. As can be seen from the volcano plots on the right side, *Cyclotella* is indeed slightly skewed towards upregulated genes, but the majority of *Cyclotella* genes (about 80%) are not differentially expressed.

Taken together, these results indicate that the upregulation of diatom sex markers is not a general consequence of increased diatom abundance. However, we agree that normalization by genus-level abundance would be the most rigorous approach. We have therefore repeated all analyses relative to libraries with all genus-level reads (see below).

I have seen in the revised version that the authors have added 3 vegetative genes as a control. Is it possible to add the expression of these genes in Figure 4e to show that not all diatom genes are upregulated? Another possibility could be to use all *Cyclotella* transcripts as a baseline and see if *Cyclotella* sex gene markers represent a higher proportion of its transcripts in the salt condition (and do the same for each diatom genus).

As suggested by the Reviewer, we subdivided the microcosm RNA-seq dataset into the four *Thalassiosirales* clades (*Cyclotella*, *Thalassiosira*, *Conticribra* and *Skeletonema*) using the new Kraken2 taxonomy. By performing a separate differential expression analysis for each of these clades, we corrected for differences in library size (total number of reads captured) due to changes in relative abundance for that specific taxon. The results of these statistical tests were nearly identical to the DE analysis over all taxa we performed earlier (26/27 sex markers identical).

We also recalculated the counts per million expression values for each clade separately, and this also did not change the outcome (see comparison below). We have updated the manuscript, figures and Zenodo datasets with these new per-clade normalized DE analysis and figures.

I don't understand what is the conclusion of Figure 4f. As *Thalassiosira* is less abundant than *Cyclotella*, all *Thalassiosira* genes will have a lower read count. I do not see anything specific to the sexual gene markers here (except if I missed something in the normalization method).

We agree that this panel was mainly showing differences in abundance of the individual species and so is not really informative. We have removed it in favor of the barplot of 18S barcoding (suggested below).

Thalassiosira and *Conticribra* do not appear in the 18S barplot (Figure 4c) because of their low abundance. Is it possible to add a panel showing the variation of the 18S relative abundance of the four clades highlighted in panels e and f ?

We added barplots showing the 18S relative abundance of the four main *Thalassiosirales* clades during the microcosm experiment as a new Figure 4d.

The methodology employ to find sex genes in Tara Oceans MAGs is good and I'm convinced by the expression of diatom sex genes in many oceanic regions (Figure 5). The use of imaging data is a nice complement to prove the presence of sexual events.

However, I'm not convinced by the conclusion that "there is more abundance of *Chaetoceros* in stations where there is a sexual activity" (lines 302-306). How can we exclude the possibility that sex genes are detected only because *Chaetoceros* is abundant enough to exceed the sequencing depth threshold? Are there oceanic stations with high abundance of *Chaetoceros* without the expression of sex genes ? If so, this could be very interesting and should be presented. Otherwise, we could interpret the results as sex genes are expressed at low level everywhere and only when the species is abundant enough we can detect it within metatranscriptome data.

We thank the reviewer for this insightful comment. We agree that it is important to distinguish between biological regulation and detection limits driven by species abundance. First, we would like to clarify that we apply a normalization strategy that mitigates the confounding effect of species abundance. Specifically, we compute MAG-level TPM values, meaning that the expression of each gene is normalized to the total expression from the same MAG. This within-MAG normalization ensures that genes whose expression scales linearly with MAG abundance do not increase in TPM as the species become more abundant (e.g. *Chaetoceros*). Therefore, our fixed TPM expression thresholds for detecting sexual reproduction (based on experimental data) are independent of whether a species is rare or abundant in a sample.

Of course, as suggested by the Reviewer, it is possible that sexual reproduction is more widespread than detected, and that sex markers are missed in some samples due to insufficient sequencing depth. However, we note that we have successfully detected sex marker expression in several diatom MAGs that are locally very rare (e.g., <1% of mapped reads), suggesting that sex signals can be detected even when abundance is low.

As for *Chaetoceros*: this genus indeed shows significantly higher abundance in stations where sexual reproduction signals are detected, as shown in our statistical analysis (lines 302-306). However, we have also observed several stations with high *Chaetoceros* abundance but no evidence of sex marker expression (see Supplementary Figure 28 and additional bar plots provided below). This suggests that high abundance alone is not sufficient to explain the presence of sex marker expression.

To acknowledge the valid concern raised by the reviewer, we now include a sentence in the main text to acknowledge that abundance-driven detection cannot be fully excluded, but it is unlikely to be the only explanation, as sexual reproduction signals are not detected in all high-abundance stations.

Rebuttal Figure. Relative abundance of *Chaetoceros* (size fraction: micro, i.e.: 20-180 μ m) across Tara Oceans stations, grouped by 18S region (V4 or V9) and depth layer (SRF: Surface, or DCM: Deep Chlorophyll Maximum). Bars are color-coded by the presence (turquoise) or absence (red) of *Chaetoceros*-specific sex marker gene expression (*sex_signal*).

In the discussion (lines 368-369), to say that sexual reproduction is more prevalent in some stations, I think the expression of sexual genes needs to be normalized to the expression of all other genes for each diatom species. Same problem for *Thalassiosira* in station 123 (lines 380-382).

We thank the reviewer for this thoughtful comment. We agree that normalization is critical to draw meaningful conclusions about gene expression levels. As correctly pointed out, the

expression of sex marker genes must be interpreted in the context of the overall transcriptional activity of the corresponding diatom species.

As explained above, we calculated MAG-level TPM values, meaning that each gene's expression was normalized by the total number of reads mapped to the same MAG in each sample. This ensures that the expression of sex marker genes is normalized relative to the overall transcriptional activity of the same MAG, effectively correcting for both transcript abundance and the underlying MAG abundance across samples. This normalization allows for robust comparisons across stations.

To clarify this methodological point, we revised the manuscript as follows:

Revised text:

“By calculating MAG-level TPM values, we correct for the relative abundance of each MAG across samples, ensuring that the expression of sex marker genes is normalized relative to the overall transcriptional activity of the same MAG. This enables consistent comparisons across stations and provides a robust basis for applying gene-specific expression thresholds determined for single species in a laboratory setting.”

Is it possible to quantify the proportion of cell ongoing sexual reproduction with imaging datasets ? I think this would be a good indication of a higher prevalence in a sample.

Quantification of sexual and vegetative cell stages in the IFCB/EcoTaxa data is not possible because of limitations of this dataset, in particular the variability in data quality between samples. Some stations have thousands of pictures, others only have a few or captured just air bubbles and no cells.

However, we previously performed a qualitative survey of the IFCB dataset: for each station, we recorded whether we observed vegetative cells and/or sexual cell stages for each genus, and cross-referenced this with the sex marker signal. This analysis also supports that sex is not always occurring when a genus is abundant: we often identified vegetative cells of genera that did not display a sex signal (metatranscriptome) and sexual cell stages (IFCB).

This dataset can be found in our Zenodo repository, or through this link: https://zenodo.org/records/14963127/files/Ecotaxa_IFCB_observation_sheet.xlsx?download=1 . We have more clearly referred to this in the context of *Chaetoceros* abundance.

Other points :

Figure 1c : Statistical tests on Pearson correlations are missing.

Thank you for spotting this. We added a matrix with adjusted p-values of statistical testing on Pearson correlations to Fig. S1 and discuss the results in the main text.

Line 654 : Could you specify which metatranscriptomic samples you used for the assembly ? I assume a co-assembly of the 6 samples ?

Yes, that is correct, we have now clarified this in the Methods section.

Line 675-676 : Did you apply EdgeR differential expression directly to the 850,000 transcripts ? I would be curious to see the MA plot or violin plot (coloured by taxa).

Originally, we had indeed applied EdgeR to the entire dataset of >800.000 transcripts. You can find volcano plots from this statistical test, separated for the four taxa that we focus on, pasted at the beginning of this rebuttal.

Note that based on your feedback, we have now changed the procedure and performed differential expression analyses separately for each *Thalassiosirales* clade (hence taking into account differences in library size due to relative abundance in each sample).

Figure 4e (and related SupFigures) : Could you explain how this normalisation between 0 and 1 was done ? Is it a simple ratio per gene of the CPM values ? I think it is also important to see the CPM counts here (or at least the average per line).

Indeed, the “normalized CPM” we use for heatmaps are CPMs that are scaled to a maximum of one per gene. This is now explained more explicitly in the captions.

We used gene-level scaling because the genes with the highest expression during sex made it impossible to see the expression dynamics for other genes in the heatmap. E.g. *Sig1* in *Conticribr*a reaches about 2500 CPM during sex (0.25% of the entire transcriptome) which makes other genes nearly invisible when using a single non-normalized scale. What’s more, we often identified multiple isoforms (or duplicates) of the same marker in the same taxon, and while all of them were upregulated after the salinity shift, their baseline expression differed substantially. Because we wanted to visualize that each marker hit (transcript, isoform) in itself is specific for sex, we used the normalization so that also the lower expressed ones are visible.

We agree that it is more fair to also show the raw (non-scaled) CPM. Because the heatmaps could not clearly display this, we made separate barplots with raw CPMs for all marker hits in Fig. 4f, and included them as a new Supplementary Figure (S30).

Lines 721 to 728 : These phylogenies used to select marker genes within Tara MAGs should be

available somewhere to understand how the MAG proteins were selected according to the phylogenetic placement (as SupFigure or in a public repository).

We completely agree that the phylogenetic selection of orthologs of sex markers should be transparent. To this end, the phylogenetic trees are available on the public Zenodo repository, both for the microcosm and the Tara Oceans phylogenetic analyses. In fact, these trees were already available on Zenodo for the previous versions of this manuscript (<https://zenodo.org/records/14963127>, under “Phylogenetic_selection.zip”). However, this wasn’t clearly stated in the data availability section. We now clarified this.

Line 359-360 : “more frequently in the global ocean“ is not clear for me. Do you mean “higher frequency of sexual events in natural diatom populations than previously though” ? I think this has not been shown in the manuscript. Instead, you could write “sexual events occur in more oceanic regions than previously though”.

We thank the reviewer for this helpful comment. We agree that our original wording may have implied a quantitative increase in the frequency of sexual events, which we do not directly measure. Our intended message is that because diatom sex is traditionally considered rare in nature, the detection of co-expression of sex marker genes across more than 50 Tara Oceans stations is already noteworthy. While we do not quantify frequency, the widespread geographic occurrence of this signal suggests that sexual reproduction may be more common than previously assumed.

To reflect this more clearly, we have revised the sentence as follows:

Revised text (lines 358-360):

Although the co-expression of sex markers only serves as an indirect proxy for sexual activity, the identification of a sexual signal across more than 50 *Tara* Oceans stations worldwide is notable, given the long-standing view that diatom sex is a rare event in nature

To ensure consistency, we also revised the corresponding sense in the abstract:

Revised abstract sentence:

“Our results reveal that diatom sexual reproduction is more widespread in the global ocean than previously thought, encompassing both dominant bloom-forming species and rare taxa.”